# Phase separation of Polo-like kinase 4 by autoactivation and clustering drives centriole biogenesis

Jung-Eun Park [1], Liang Zhang [1], Jeong Kyu Bang [2], Thorkell Andresson[3], Frank DiMaio[4] & Kyung S. Lee [1]*

Tight control of centriole duplication is critical for normal chromosome segregation and the maintenance of genomic stability. Polo-like kinase 4 (Plk4) is a key regulator of centriole biogenesis. How Plk4 dynamically promotes its symmetry-breaking relocalization and achieves its procentriole-assembly state remains unknown. Here we show that Plk4 is a unique kinase that utilizes its autophosphorylated noncatalytic cryptic polo-box (CPB) to phase separate and generate a nanoscale spherical condensate. Analyses of the crystal structure of a phospho-mimicking, condensation-proficient CPB mutant reveal that a disordered loop at the CPB PB2-tip region is critically required for Plk4 to generate condensates and induce procentriole assembly. CPB phosphorylation also promotes Plk4's dissociation from the Cep152 tether while binding to downstream STIL, thus allowing Plk4 condensate to serve as an assembling body for centriole biogenesis. This study uncovers the mechanism underlying Plk4 activation and may offer strategies for anti-Plk4 intervention against genomic instability and cancer.

---

[1] Laboratory of Metabolism, National Cancer Institute, National Institutes of Health, 9000 Rockville Pike, Bethesda, MD 20892, USA. [2] Division of Magnetic Resonance, Korea Basic Science Institute, 162 Yeongudanji-ro, Ochang-eup, Cheongju 28119, Republic of Korea. [3] Protein Characterization Laboratory, Frederick National Laboratory for Cancer Research and Leidos Biomedical Research Inc., 8560 Progress Drive, Frederick, MD 21702, USA. [4] Department of Biochemistry and Institute for Protein Design, University of Washington, 1705 NE Pacific Street, Seattle, WA 98195, USA. *email: kyunglee@mail.nih.gov

As the main microtubule (MT)-organizing center in animal cells, the centrosome plays a pivotal role in various cellular processes, such as spindle formation, chromosome segregation, and cell division. The centrosome is composed of a pair of MT-derived apparatus, called centrioles, embedded in a pericentriolar protein matrix. Centriole duplication occurs precisely once per cell cycle, and tight control of centriole numbers is essential for the maintenance of genomic stability[1,2].

Centriole duplication begins by assembling a procentriole in the late G1/early S phase, and timely activation of Polo-like kinase 4 (Plk4) appears to be central for inducing centriole biogenesis[2,3]. Notably, Plk4 exhibits a ring-like localization pattern (i.e., ring state) around the Cep152 scaffold in early G1, and then subsequently adopt a dot-like morphology (i.e., dot state) at the future procentriole assembly site[4–7]. The symmetry breaking from a uniformly distributed ring state to a confined dot state is critical for Plk4 to assume a physically distinct entity specialized for centriole biogenesis.

Plk4 dimerizes through the cryptic polo box (CPB; residues 581–808)[6,8,9] of its C-terminal domain (CTD; residues 581–970) and its N-terminal kinase domain (KD)-dependent *trans*-autophosphorylation activity generates a phosphodegron at S285 and S289 to induce SCF-βTrCP/Slimb-mediated ubiquitination and its own proteasomal degradation[10–14]. This observation suggests that Plk4 must overcome its phosphodegron-mediated self-destruction threshold to trigger downstream events critical for centriole biogenesis. Other studies have shown that a dimeric form of CPB (581–808) interacts with the N-terminal region of a pericentriolar scaffold Cep152 and this interaction positions Plk4 at the outskirts of a ring-like Cep152 signal[4,6,15–17]. The PB3 (residues 880–970) of CTD has been shown to bind to the STIL coiled-coil (CC) domain[18], thus allowing Plk4's N-terminal catalytic activity to phosphorylate STIL's C-terminal STAN domain, induce the STIL–Sas6 interaction, and assemble a cartwheel-like structure[5,18–21].

It is widely accepted that intrinsically disordered regions or low-affinity multivalent interactions can promote liquid–liquid-phase separation (LLPS), providing a driving force for generating various biomolecular condensates[22,23]. A recent study shows that wild-type (WT) *Xenopus* Plk4 can self-assemble into sphere-like condensates, whereas its inactive mutant generates an amorphous network[24]. Another report suggests that human Plk4 gains a self-organizing activity by dephosphorylating a flexible linker region (residues 280–305)[25] that has been shown to function as the phosphodegron motif for βTrCP[25]. It is unclear how the dephosphorylated linker region works in concert with its N-terminal catalytic activity to form a functional Plk4 assembly.

Here we demonstrate that Plk4 promotes its own ring-to-dot localization conversion by autophosphorylating and transmuting the physicochemical properties of its noncatalytic CPB, thereby causing it to rapidly coalesce into a nanoscale spherical condensate with a distinct constituent phase. Mutations in the disordered region within CPB eliminate phospho-CPB-dependent Plk4 condensation, Plk4's symmetry-breaking ring-to-dot relocalization, and its ensuing centriole biogenesis. Thus, we propose that Plk4 is an unparalleled kinase that harnesses its KD-dependent autophosphorylation activity to trigger its CPB-dependent physicochemical condensation. This unique capacity enables Plk4 to phase separate into a matrix-like body that can amass downstream components critical for procentriole assembly.

## Results

### Plk4's ring-to-dot conversion requires CPB phosphorylation.
Using three-dimensional structured illumination microscopy (3D-SIM), we observed that treatment of cells with a Plk4 inhibitor, centrinone[26], was sufficient to prevent Plk4's ring-to-dot localization conversion, as shown previously[27], and that this event is essential for the subsequent recruitment of Sas6 to the procentriole assembly site (Supplementary Fig. 1a). In addition, overexpressed Plk4 WT, but not its catalytically inactive form, induces multiple patches of submicron-scale electron-dense material[28], suggesting that Plk4 may exhibit unusual physicochemical properties capable of forming dot-like aggregates.

Catalytic activity-dependent ring-to-dot conversion hints that Plk4 induces a symmetry-breaking process through its autophosphorylation activity. Since Plk4 is a suicidal kinase that degrades through a self-generated phosphodegron for βTrCP[12,13], it must circumvent its own destruction to trigger centriole duplication. An earlier report suggests that, when sufficiently concentrated, *Drosophila* Plk4 can promote its own activation[29]. Therefore, if the dot-state Plk4 represented physically clustered Plk4, a high level of Plk4 expression would be needed to mimic the physicochemical environment of the dot state. Overexpression of EGFP-Plk4 yielded hyperphosphorylated and slow-migrating Plk4 forms (Supplementary Fig. 1b). Mass spectrometry (MS) analysis with immunoprecipitated EGFP-Plk4 revealed multiple clustered phosphorylations within the CTD (referred to hereinafter as phosphocluster PC1–PC8) (Fig. 1a and Supplementary Fig. 1b, c). Subsequent analysis with pc mutants (all phosphosites were mutated to Ala) revealed that the pc3 mutant (S698A, S700A, T704A, T707A) (Fig. 1b and Supplementary Fig. 1d) migrated nearly as fast as the catalytically inactive K41M (KM) mutant (Supplementary Fig. 1e), suggesting a conformational change by PC3 phosphorylations.

In U2OS cells stably expressing a low level of exogenous Plk4, using a lentivirus-based system, and silenced for endogenous Plk4 by small interfering RNA (siRNA) (siPlk4), the pc3 mutant exhibited a greatly diminished ability to recruit Sas6 to centrosomes, although its catalytic activity and homodimerization ability were not noticeably altered (Supplementary Fig. 1f, g; see Supplementary Fig. 2k below). Localization analysis revealed that the pc3 mutant exhibited a high level of the ring-state Plk4, almost comparable to that of the Plk4 KM mutant (Fig. 1c), suggesting that KD-dependent phosphorylations at the PC3 motif are critical for Plk4's ability to convert from a ring state to a dot state. Notably, the PC3 residues (Supplementary Fig. 1c, right; green color), located at the PB1-PB2 bridge (S700–P701) and its adjacent PB2 β7 region, are highly conserved throughout evolution (Fig. 1b).

To confirm the importance of PC3 phosphorylation for Plk4-mediated centriole biogenesis, we generated Plk4 RNA interference (RNAi) cells expressing a Plk4 mutant lacking all PC residues (i.e., pc1–8) but bearing restored PC3 WT residues (i.e., pc1–8 + PC3). The resulting mutant effectively recruited Sas6 to centrosomes at a level comparable to that of WT cells (Fig. 1d). Notably, the pc3-mutant cells still displayed a significant level of Sas6, in comparison to that of the pc1–8-mutant cells (Fig. 1d), hinting that phosphorylations at the other seven PC residues may also contribute to this event.

### Plk4 CP requires STIL, Not Cep152, for procentriole elongation.
Next, we systematically mutated the four PC3 residues to negatively charged aspartic or glutamic acid and generated a condensation-proficient (CP) gain-of-function mutant (i.e., the S698E, S700E, T704E, T707D mutant). In U2OS cells where endogenous Plk4 is silenced, the CP mutant exhibited a greatly increased level of Sas6 recruitment to centrosomes (Supplementary Fig. 2a). A low but significant level of Sas6 was also detected in cells expressing the catalytically inactive KM CP mutant, suggesting that the CP mutations at the PC3 motif of CPB can

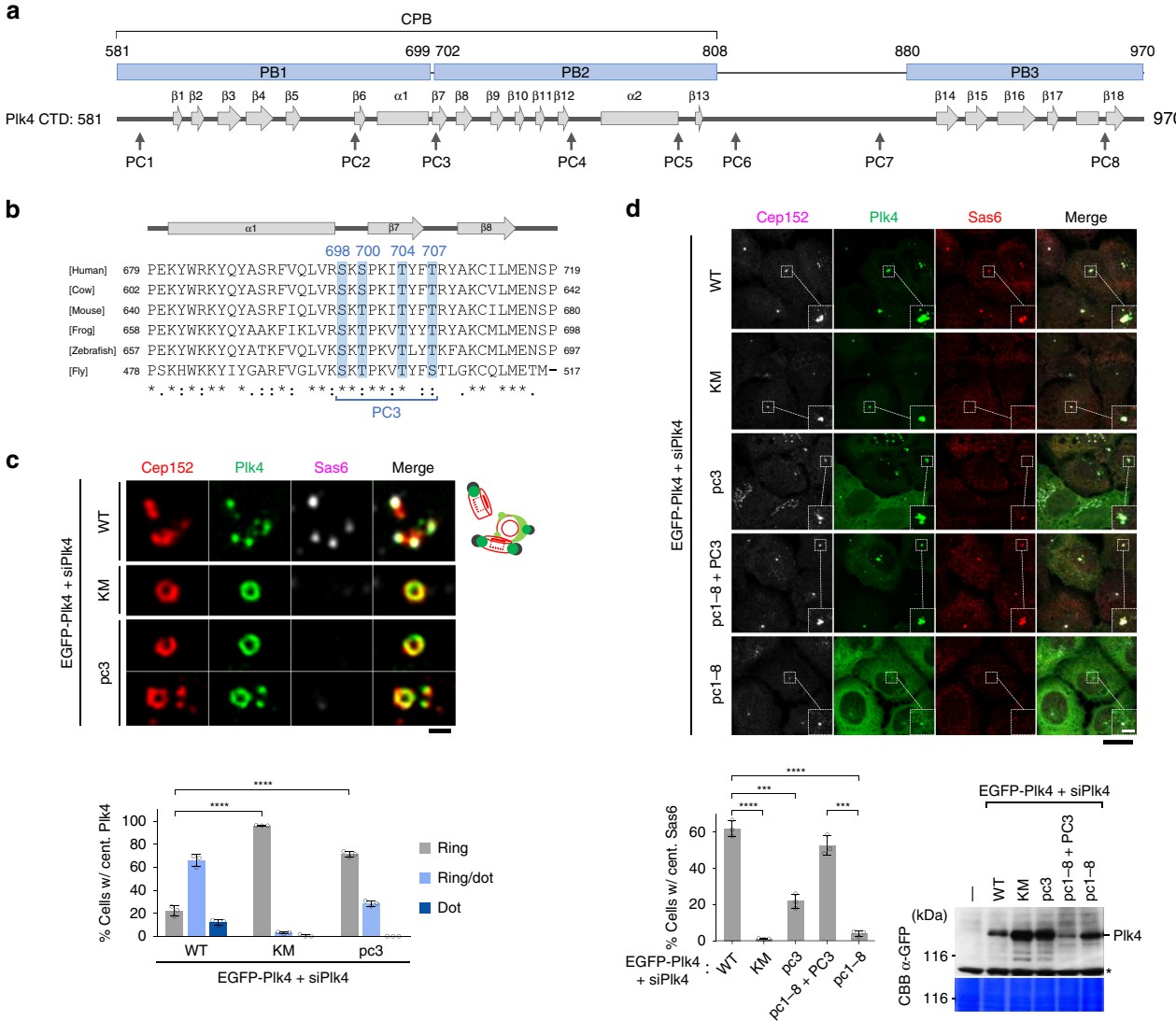

**Fig. 1** Plk4 triggers its symmetry-breaking ring-state-to-dot-state relocalization by autophosphorylating its CPB. **a** Schematic diagram showing the secondary structure of the Plk4 CTD. Numbers indicate amino acid residues. The positions of PC1 to PC8 are marked. **b** Multiple sequence alignment for the region containing PC3 was performed using the Clustal Omega software. The S698, S700, T704, and T707 residues phosphorylated in vivo are indicated. **c** 3D-SIM analysis of immunostained U2OS cells stably expressing the indicated EGFP-Plk4 constructs and silenced for endogenous Plk4 (siPlk4). The schematic diagram (right) illustrates multiple Cep152 ring (red), Plk4 dot (green), and Sas6 (black) signals. Bar, 0.5 μm. Quantification of images is shown in mean ± s.d. ($n = 3$ independent experiments). Note that like the catalytically inactive Plk4 KM mutant, the pc3 mutant remained at a ring state. **** $P < 0.0001$ (unpaired two-tailed $t$ test). **d** Confocal analysis of immunostained U2OS cells stably expressing various EGFP-Plk4 constructs under Plk4 RNAi conditions (siPlk4). Bar, 20 μm. Dotted boxes, areas of image enlargement (bar, 3 μm). Quantified data are shown in mean ± s.d. ($n = 3$ independent experiments). Immunoblotting shows the expression of each construct. ***$P < 0.001$; ****$P < 0.0001$ (unpaired two-tailed $t$ test). Asterisk denotes a cross-reacting protein. CBB, Coomassie Brilliant Blue-stained membrane. Source data are provided as a Source Data file for **c**, **d**

alleviate the essential requirement of the N-terminal catalytic activity in recruiting Sas6. Quantification carried out with cyclin A-positive G2 cells also yielded similar results (Supplementary Fig. 2b), demonstrating that increased Sas6 recruitment is not due to an altered cell cycle.

3D-SIM analyses for the cells in Supplementary Fig. 2a showed that while ~38% of Plk4 WT-expressing cells displayed multiple dot-like Plk4 signals, ~95% of the Plk4 CP cells exhibited multiple Plk4 dots with greatly enhanced Sas6 signals (Fig. 2a). The catalytically inactive KM mutant remained as a ring without any detectable Sas6. Not surprisingly, the KM CP mutant induced a significant level (32%) of multiple Plk4 dot signals that were frequently colocalized with Sas6 (Fig. 2a, panel 7 and graph).

One striking observation is that a large fraction (>70%) of the CP mutant cells exhibited multiple elongated (>250 nm) Plk4 signals colocalized with STIL and Sas6 along their entire length (Fig. 2b and Supplementary Fig. 2c). The Plk4 CP-induced dot-like or elongated signals exhibited a dynamic exchange with the bulk solution (Supplementary Fig. 2d; Supplementary Video 1), suggesting that Plk4 CP in the dot or elongated morphology is not at a solidified state. Cells expressing WT Plk4 were also capable of producing an elongated Plk4 fluorescence, but at a much reduced level (Fig. 2b and Supplementary Fig. 2c). In a second experiment, endogenous promoter-controlled Plk4 CP exhibited an increased capacity to recruit Sas6 and induced an elongated procentriole-like morphology (Supplementary Fig. 2e–g), albeit at a lesser degree than the cytomegalovirus (CMV) promoter-controlled Plk4 CP (Fig. 2a, b).

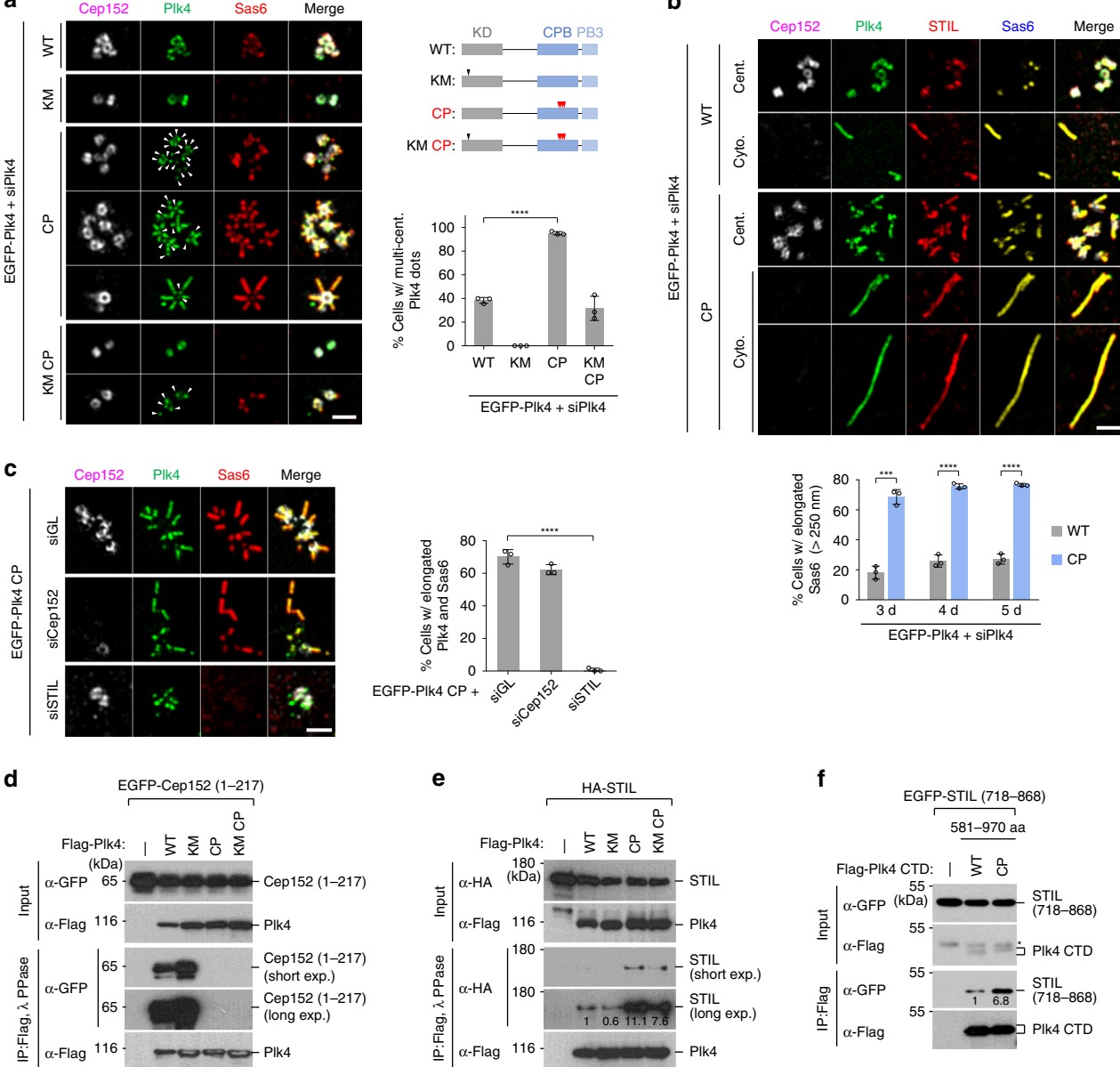

**Fig. 2** A condensation-proficient Plk4 CP mutant drives procentriole formation by switching its interaction from Cep152 to STIL. **a, b** 3D-SIM analysis of immunostained U2OS cells stably expressing the indicated EGFP-Plk4 constructs under Plk4 RNAi (siPlk4) conditions. Bar, 1 μm. Quantification of images is shown in mean ± s.d. ($n = 3$ independent experiments). Arrowheads in **a**, dot-state Plk4 signals; schematic diagram in **a** shows the sites of the KM and the quadruple CP mutations. Elongated procentrioles in both centrosome (cent.) and cytosol (cyto.; as judged by the absence of the centrosomal Cep152 signal) are shown in **b**. ***$P < 0.001$; ****$P < 0.0001$ (unpaired two-tailed $t$ test). **c** 3D-SIM analysis of immunostained U2OS cells stably expressing the EGFP-Plk4 CP mutant under various RNAi conditions. Bar, 1 μm. Quantified data are shown in mean ± s.d. ($n = 3$ independent experiments). ****$P < 0.0001$ (unpaired two-tailed $t$ test). **d–f** Immunoprecipitation (IP) and immunoblotting analyses using HEK293T cells cotransfected with the indicated constructs. IP samples were then treated with λ phosphatase (PPase), where indicated, to convert phosphorylated, slow-migrating forms into a fast-migrating form, and then separated by 8% sodium dodecyl sulfate–polyacrylamide gel electrophoresis (SDS-PAGE) for immunoblotting. Numbers denote relative signal intensities. Asterisk denotes remnants of STIL signals from a prior anti-GFP blotting. Source data are provided as a Source Data file for **a–c**

Although Plk4 CP's 2-fold-increased expression level (Supplementary Fig. 2e)—which likely stems from its ability to evade βTrCP-dependent proteasomal degradation (see below)—may have contributed to its capacity to induce centriole biogenesis, given that Plk4 CP generated cytosolic assemblies even in the absence of Cep152 (Fig. 2b and Supplementary Fig. 2c), Plk4 CP appeared to potentiate centriole biogenesis under physiological conditions. Subsequent analysis showed that while depletion of Cep152 did not significantly diminish the Plk4 CP-induced elongated signal, STIL was required for this event (Fig. 2c).

**CP mutations promote STIL binding but disrupt Cep152 binding.** Cep152-independent but STIL-dependent elongated CP fluorescence suggests that the CP mutant may have an altered binding capacity to Cep152 and STIL. Indeed, Plk4 WT or KM efficiently coprecipitated the N-terminal Cep152 (1–217) fragment, the region that interacts with CPB[4], or the full-length Cep152, whereas their respective CP and KM CP mutants did not (Fig. 2d and Supplementary Fig. 2h). Interestingly, however, the CP mutations greatly enhanced the ability of Plk4 WT or KM to bind to STIL (Fig. 2e). An experiment carried out with Plk4 CTD

(581–970) and the STIL (718–868) CC fragment containing the Plk4 PB3 (880–970)-binding region[18] also yielded a similar result (Fig. 2f). Given that STIL binds to the C-terminal PB3 motif of Plk4 CTD[18], the CP mutations in CPB may have altered the conformation of the CTD in such a way that its PB3 motif becomes accessible for STIL binding.

In a related experiment, immunoprecipitation (IP) performed with a mixture of individually transfected lysates showed that, unlike Plk4 WT or KM, the Plk4 CP mutant coprecipitated STIL but not Cep152 (Supplementary Fig. 2i). Under cotransfected conditions, the CP mutations promoted the kinase-inactive Plk4 KM-STIL interaction and Plk4's catalytic activity further enhanced this interaction (Supplementary Fig. 2j). The CP mutations did not alter the homodimerization activity of CPB (Supplementary Fig. 2k). Thus, phosphorylating the PC3 motif (Fig. 1) is critical for Plk4 to switch its binding target from Cep152 to STIL and to induce centriole biogenesis.

**Plk4 CP condensates localize along elongated procentrioles**. To investigate the morphological nature of the dot-state Plk4, we took advantage of the U2OS cells stably expressing EGFP-Plk4 CP that shows multiple dot-like and elongated fluorescent signals (Fig. 2a, b). Thin-section transmission electron microscopy (TEM) revealed that these cells exhibit multiple centrioles, frequently with a visibly elongated MT morphology (Supplementary Fig. 3a, b). Additional TEM analyses revealed spherical structures that ranged from ~60–140 nm in diameter (Supplementary Fig. 3c). Immunoelectron microscopy confirmed that these spherical assemblies are made of Plk4 and, likely, its associated proteins (Fig. 3a). Consistent with the data shown in Fig. 2b and Supplementary Fig. 2c, these assemblies were also found in the cytosol (Fig. 3a, bottom). However, they exhibited no structurally distinguishable morphology, hinting that Plk4 CP molecules may have coalesced into electron-dense condensates (see below) without an architectural rule of assembly. Furthermore, while most of the condensates with smaller diameters appeared to be solid spheres, larger condensates were found to be hollow (Supplementary Fig. 3c). Thus, we postulate that spherical Plk4 condensates could be an amorphous matrix-like body that can not only recruit downstream components but also foster the interactions among them to promote procentriole assembly.

Further analysis of Plk4 CP condensates with correlative light and electron microscopy (CLEM) revealed long electron-dense structures that closely correlated with elongated Plk4 signals along their entire length (Fig. 3b and Supplementary Fig. 3d). However, like the spherical condensates in Fig. 3a, no obvious structural arrangement was apparent. Measurement of the dimension of these structures showed that they are ~140 nm thick and frequently longer than 1000 nm in length. Considering that the outer diameter of the centriolar MT triplets is ~230 nm (Fig. 3a), the long electron-dense structure may represent procentriolar lumenal materials, likely containing Plk4 CP and its associated proteins (e.g., STIL and Sas6).

In a related 3D-SIM analysis, we also observed that, while most of the elongated Plk4 fluorescence was indistinguishably colocalized with elongated Sas6 signals, a low but significant level (2.5% ± 0.4%) of elongated Plk4 signals were found at the outskirts of the elongated Sas6 signals (Fig. 3c; Supplementary Video 2). The average peak-to-peak distance across tubular Plk4 signals was estimated to be ~136 ± 5 nm ($n = 13$) (Fig. 3c), which is significantly wider than the diameter (~90 nm) of the C terminus of Sas6 in human cartwheel structure[30]. Since the inner diameter of centriolar MT triplets in human cells is ~160 nm[31], Plk4 is likely located at the periphery of centriole lumen.

**CP mutations enhance Plk4 stability and downstream signaling**. The finding that the Plk4 CP can generate spherical condensates suggests that an active Plk4 could be at a self-interactive state that may alter Plk4's susceptibility to βTrCP-mediated proteasomal degradation. As reported previously[10–14], we observed that the level of Plk4 WT, but not the catalytically inactive Plk4 KM, was noticeably increased in cells depleted of endogenous βTrCP by siRNA (siβTrCP) (Fig. 4a). Surprisingly, although Plk4 CP is catalytically active with a potent procentriole assembly activity (Fig. 2a, b), the level of the Plk4 CP mutant remained unchanged by βTrCP depletion. The stability of Plk4 CP was comparable to that for the KM mutant under the conditions where protein synthesis is inhibited by cycloheximide (Fig. 4a). Consistently, while the signal intensities of centrosome-localized Plk4 WT were greatly increased (>2-fold) by βTrCP depletion, those of Plk4 KM and Plk4 CP were not (Supplementary Fig. 4a). In a related experiment, unlike ring-shaped Plk4 WT (Fig. 4b, yellow arrow), dot-shaped or elongated Plk4 WT and CP were devoid of colocalized βTrCP signals (Fig. 4b, arrowheads and dotted boxes). Thus, PC3 phosphorylation-induced condensation ensures Plk4 stability by evading βTrCP binding.

The formation of a condensate could be disadvantageous unless an enzyme can freely interact with its recruited substrates. Therefore, we examined whether Plk4 CP's ability to potently induce procentriole-like structure is indeed due to its increased capacity to phosphorylate the STIL STAN motif by using an antibody generated against S1108-phosphorylated STAN (i.e., pS1108) (Supplementary Fig. 4b). A Plk4 pSSTT antibody, which detected phospho-PC3 residues (i.e., pSSTT) (Supplementary Fig. 4c, d) and a dot-state Plk4 in a kinase activity-dependent manner (Fig. 4c and Supplementary Fig. 4e), also efficiently decorated the CP mutation-containing epitope and therefore was included for analysis. As expected, cells expressing Plk4 WT, but not its respective KM mutant, exhibited readily discernable fluorescent signals for Plk4 pSSTT, STIL, STIL pS1108, and Sas6 at centrosomes (Fig. 4d and Supplementary Fig. 4f–h). Under these conditions, Plk4 CP-expressing cells displayed significantly (2.25-fold) increased STIL fluorescence with proportionally increased STIL pS1108 and Sas6 signals at centrosomes. This suggests that an increased ability of Plk4 CP to recruit STIL is accountable for the increase in Plk4-dependent STAN phosphorylation and Sas6 recruitment. As expected, pSSTT and pS1108 epitopes and Sas6 signals were also manifest along the entire length of Plk4 CP-induced elongated procentriole-like structure (Fig. 4e). These findings suggest that Plk4 in the dot or elongated state can effectively phosphorylate STIL and recruit Sas6, and that phospho-PC3 Plk4, pS1108 STIL, and Sas6 are the constituents of the electron-dense matrix-like bodies shown in Fig. 3a, b.

**CP mutations induce flexibility at the CPB PB2-tip region**. To understand the underlying mechanism of how PC3-phosphorylated CPB can drive Plk4 to coalesce into a stable dot-like condensate, we examined whether CPB CP contains a flexible region that may cause an increase in accessible hydrophobicity, a physicochemical property known to be critical for driving protein–protein interactions[32,33]. 1-Anilinonaphthalene-8-sulfonic acid (ANS) alters its fluorescent properties as it binds to nonpolar regions and therefore is considered an effective probe to investigate the remodeling in the accessible hydrophobic surfaces of a protein[34,35]. Interestingly, while WT CPB exhibited only a low level of ANS binding, the CPB CP mutant displayed drastically increased ANS binding at an elevated temperature (Fig. 5a). This suggests that the conformation of CPB CP renders a temperature-dependent dynamic disorder and that CPB CP could be condensation-prone

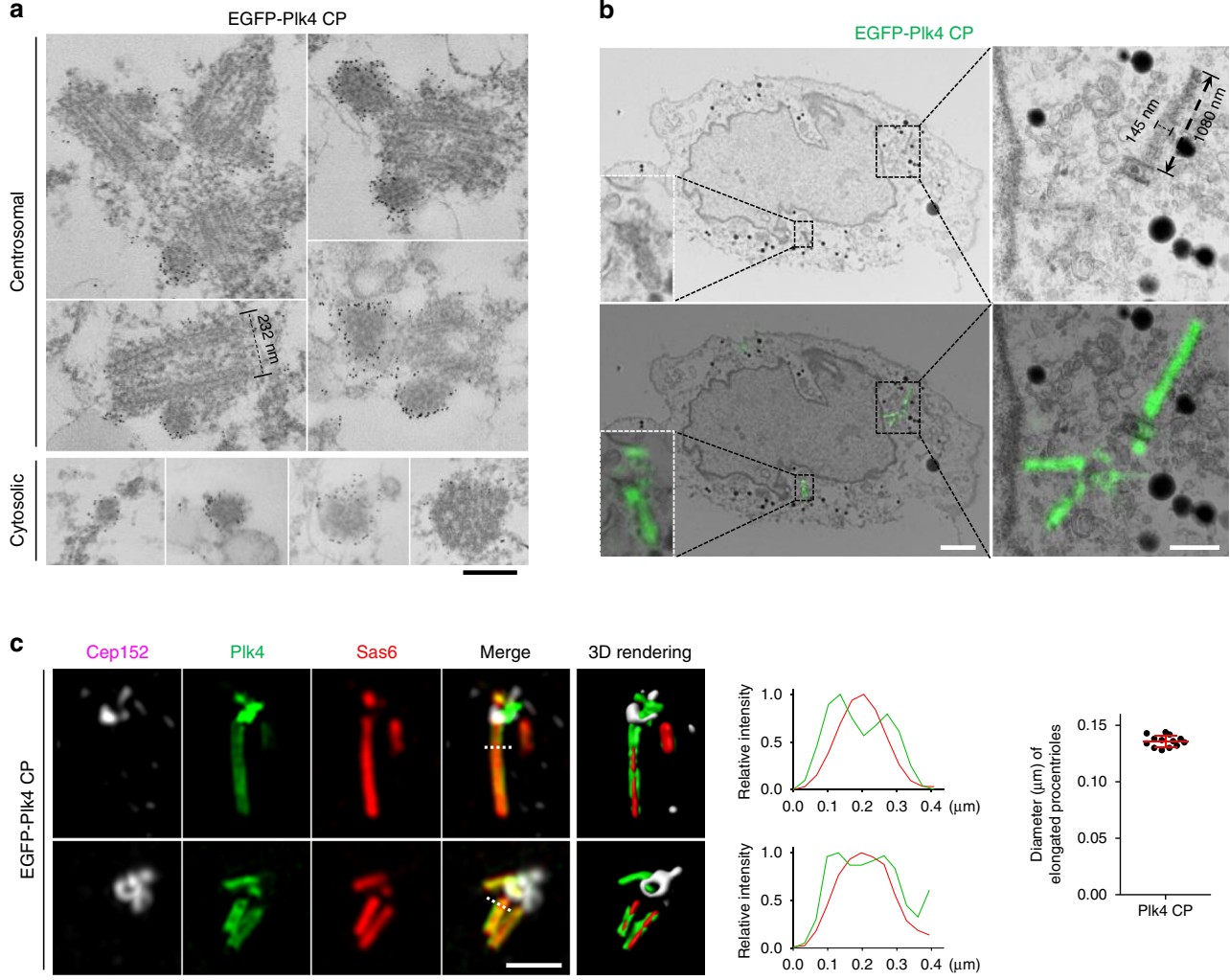

**Fig. 3** A condensation-proficient Plk4 CP mutant generates nanoscale, electron-dense bodies and localizes along an elongated procentriole. **a** Thin-section transmission electron microscopy (TEM) showing U2OS cells stably expressing EGFP-Plk4 CP stained with an anti-GFP and a 6-nm nanogold-conjugated secondary antibody. Both centrosomal and cytosolic Plk4 condensates decorated with nanogold signals are shown. Bar, 0.2 μm. **b** Correlative light and electron microscopy (CLEM) analysis of U2OS cells stably expressing EGFP-Plk4 CP. Electron micrographs overlaid with GFP fluorescent 3D-SIM images are shown at the bottom. Bar, 2 μm. Dotted boxes, areas of enlargement. Bar, 0.5 μm. The dimension of an elongated electron-dense body induced by Plk4 CP is shown. **c** Two 3D-SIM images showing tubular EGFP-Plk4 signals observed in U2OS cells. Plk4 and Sas6 signals were detected along the length of elongated procentrioles. The images were subjected to 3D surface rendering (see Supplemental Movie S1). Bar, 1 μm. Dotted lines traversing the elongated procentrioles mark the region where the relative fluorescence intensities for Plk4 and Sas6 were measured (middle). Bars in the graph (right), an average peak-to-peak distance across the tubular EGFP-Plk4 signals (n = 13) with s.d. Source data are provided as a Source Data file for **c**

due to increased hydrophobic surfaces. A serendipitously generated CPB CP_v1 mutant containing a C-terminal random 32-mer extension also interacted with ANS in a similar fashion (Fig. 5a). The 32-mer extension did not alter the dimerization capacity of CPB CP or the ability of Plk4 CP to dissociate from Cep152, recruit Sas6 to centrosomes, and induce multiple elongated Sas6 signals (Supplementary Fig. 5a–d).

To gain insight into how the CP mutations alter the physicochemical properties of CPB, we determined the crystal structures of both CPB CP and CPB CP_v1. While extensive efforts to obtain a high-resolution CPB CP structure yielded only a 3.7 Å overall resolution, the structure of CPB CP_v1 was solved at 2.64 Å (Table 1). Therefore, the structure model of CPB CP was generated using a combination of RosettaCM and Rosetta/Phenix crystallographic refinement (TFZ = 16.4 for the initial model building, and $R_{\text{work}}/R_{\text{free}} = 0.2709/0.3092$ for the final model; see Methods for details). The overall structure of both CP forms was similar to that of the X-shaped, homodimeric apo-CPB (4N9J)[6],

as confirmed by TEM (Fig. 5b and Supplementary Fig. 5e, f). No densities for the C-terminal 32-mer extension of CPB CP_v1 were detected. An individual alignment of PB1 or PB2 of the CPB CP_v1 structure with the respective domain of CPB CP showed a Cα root-mean-square deviation value equal to 0.6970 or 0.7679 Å, respectively, well within the acceptable range of two super-imposable structure models. Similarities in their crystal structures and ANS-accessible hydrophobic surfaces suggest that the C-terminal 32-mer extension did not significantly alter the conformation of CPB CP.

Apo-CPB contains a stem-like, 14-stranded, antiparallel β-sheet along the length of the PB2–PB2 axis that sharply twists counterclockwise by 158°[6] (Fig. 5b, c). Remarkably, CPB CP_v1 exhibited a substantially reduced level of twisting (119°), thus forming a greatly flattened PB2–PB2 axis (Fig. 5b, c). Since twisting of the β-sheet occurs as a cumulative outcome of an inherent property of the peptide backbone in each strand, the CP mutations near the PB2 base might have generated a long-

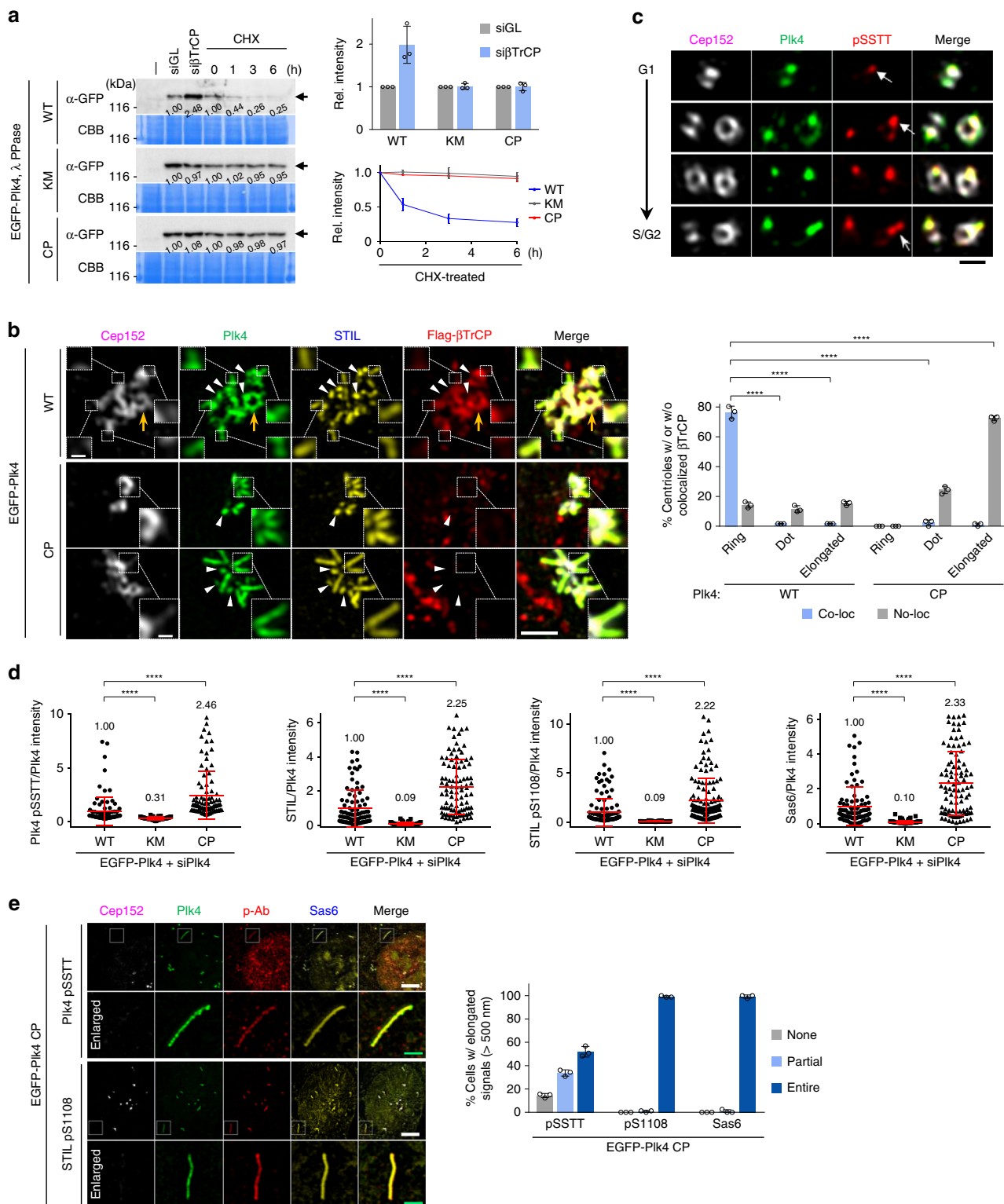

distance effect along the PB2–PB2 axis, yielding largely disordered PB2-tip residues (Fig. 5c, right; see below).

Examination of the four CP mutant residues (S698E, S700E, T704E, and T707D) revealed that each was engaged in additional interactions not found in the apo-CPB (Fig. 5d and Supplementary Fig. 5g). While E698 appeared to reinforce its interaction with Q694, both E700 and E704 were involved in rigidifying the CPB CP structure by forming extra intra-subunit interactions with K600 and K711, respectively. D707 seemed to stiffen the

CPB architecture by engaging in two interlocking interactions—a salt bridge with intra-subunit H771 and the other H-bonding with inter-subunit A796—thus securing the PB2 β7–β8 of one subunit to the CPB α2 backbone of the other subunit. Since all four CPB CP mutations were required to generate a centriole duplication-proficient Plk4, they may function in a concerted manner to adopt a condensation-prone CPB structure. Superimposing the structure of CPB CP with that of the CPB–Cep152 60-mer complex (4N7V)[6] revealed that CPB CP may fail to

**Fig. 4** Plk4 containing the phospho-PC3 motif is stable and colocalizes with STIL and Sas6 along the length of an elongated procentriole. **a** Immunoblotting analyses of U2OS cells stably expressing the indicated constructs treated with siRNA (siGL or siβTrCP) or cycloheximide (CHX). Total lysates were treated with λ phosphatase (PPase) to convert all phosphorylated, slow-migrating Plk4 forms into a fast-migrating form (arrows). Numbers, relative signal intensities. Quantification of relative signal intensities is shown in mean ± s.d. ($n = 3$ independent experiments). **b** 3D-SIM analysis and quantification of immunostained HEK293T cells stably expressing the indicated EGFP-Plk4 constructs and transfected with Flag-βTrCP. Bar, 1 μm. Enlarged images (bar, 0.2 μm) for dotted boxes are shown. The fraction of cells with or without colocalized Flag-βTrCP signals was quantified (right) (mean ± s.d., $n = 3$ independent experiments). ****$P < 0.0001$ (unpaired two-tailed $t$ test). Note that elongated (inside dotted boxes) and dot-state (arrowheads) Plk4 signals are colocalized with STIL, but not with βTrCP, while the ring-state Plk4 (yellow arrow) is colocalized with Cep152 and βTrCP. **c** 3D-SIM analysis of U2OS cells immunostained with anti-Cep152, anti-Plk4, and Alexa Fluor 594-conjugated anti-Plk4 pSSTT antibodies. Images representing different stages from G1 to S/G2 are shown. Bar, 0.5 μm. See the confocal images in Supplementary Fig. 4e that demonstrate the specificity of the Alexa Fluor 594-conjugated Plk4 pSSTT antibody. Note that the Plk4 pSSTT epitope preferentially associates with the dot state (arrows) and elongated (barbed arrow) Plk4 signals rarely observed at native centrosomes. **d** Quantification of the relative intensities of Plk4 pSSTT, STIL, STIL pS1108, and Sas6 signals was carried out using confocal images from immunostained U2OS cells. Relative fluorescence intensities (mean ± s.d.) are shown. $n = 90$ (WT), 90 (KM), and 90 (CP) for Plk4 pSSTT/Plk4, $n = 90$ (WT), 91 (KM), and 97 (CP) for STIL/Plk4, $n = 132$ (WT), 103 (KM), and 142 (CP) for STIL pS1108/Plk4, and $n = 90$ (WT), 91 (KM), and 97 (CP) for Sas6/Plk4 with each sample pooled from three independent experiments. Numbers denote mean relative intensities. Representative images are shown in Supplementary Fig. 4f–h. ****$P < 0.0001$ (unpaired two-tailed $t$ test). Numbers denote mean values. **e** 3D-SIM analysis of immunostained U2OS cells stably expressing EGFP-Plk4 CP. Bars, 5 μm. Enlarged images (green bars, 1 μm) for dotted boxes are shown. Quantification of images is shown in mean ± s.d. ($n = 3$ independent experiments). Source data are provided as a Source Data file for **a**, **b**, **d**, **e**

establish multiple key interactions with the Cep152 60mer (Fig. 5e), thus explaining why CPB CP did not interact with Cep152 (Fig. 2d).

Since dynamic disorder is considered the major contributor to the *B*-factor of a protein crystal structure[36,37], we carried out comparative *B*-factor analyses and found several disordered regions in CPB CP_v1 (Supplementary Fig. 5h, i). Sequence analysis using the PONDR software (http://www.pondr.com/) revealed that, unlike the PB1 wing edge region, the PB2 tip has strong potential to show structural disorder (Supplementary Fig. 5j). Notably, the PB2-tip region is composed of a stretch of mainly hydrophilic residues with four conserved hydrophobic residues (i.e., Y750, L752, V758, and L761) scattered along the disordered region (Supplementary Fig. 5k). A large fraction of these residues, including the four hydrophobic residues, were absent or lacking their side chains in the CPB CP structure (Fig. 5c), a striking deviation from an α-helix-containing ordered loop found in apo-CPB. Therefore, we focused on the PB2-tip region for further analysis. *B*-factor-based structural analysis has served as a measure of predicting residue flexibility[36,37] and is used to identify the region of conformational plasticity in polo-like kinase 1[38].

**PB2-tip-dependent phase separation of phospho-PC3-bearing CPB.** Consistent with the temperature-dependent ANS accessibility (Fig. 5a), CPB CP but not WT CPB became rapidly clustered after shifting the temperature from 4 to 20 °C, indicating that the formation of CPB CP condensates is thermally triggered (see below). Measurement of turbidity changes at 350 nm ($OD_{350 nm}$) showed that as little as 1–2 μM of CPB CP generated a detectable level of condensates, which were increased as a function of concentration (Fig. 6a). The kinetics of forming these condensates was very rapid, reaching a plateau within a few minutes after raising the temperature to 20 °C.

Visualization of Plk4 CPB condensates with fluorescein isothiocyanate (FITC) revealed that the CP mutant efficiently generated submicron-scale spherical condensates that frequently appeared to be interconnected (Fig. 6b, d and Supplementary Fig. 6f, below). The diameter and 3D volume of these condensates were largely proportional to the protein concentration (Fig. 6b, bottom). Since Plk4 is thought to function as a homeostatic clock that sets centriole size[39], the total amount of CPB PC3-phosphorylated Plk4 and its ensuing condensation activity could influence the length of centrioles. Time-lapse microscopic

analysis using FITC–CPB CP showed that green fluorescent condensates emerged quickly (i.e., <1 s) in a stochastic fashion and largely maintained their diameters during the course of the imaging (Supplementary Fig. 6a). This rapid condensation kinetics is not surprising because hydrophobic interaction is highly cooperative due to the collective action of water networks and occurs at microsecond timescale[40]. Like the spherical Plk4 condensates observed in U2OS cells (Fig. 3), electron microscopy analyses failed to reveal any discernable morphologies inside the CPB CP condensates (Supplementary Fig. 6b, c).

Next, we investigated whether the flexible PB2-tip region plays a role in CPB CP-induced clustering. Since hydrophobic residues have a strong disposition to cluster in an aqueous environment[32,33], we mutated the four conserved hydrophobic residues (Y750, L752, V758, and L761) present at the disordered PB2-tip region (Supplementary Fig. 5h–k) to Ala and examined their effect. The resulting CP PB2-tip mutant exhibited all the biochemical characteristics of the activated CP mutant, as judged by its normal dimerization activity, enhanced binding to STIL, and unbinding from Cep152 (Supplementary Fig. 6d, e). Strikingly, unlike CPB CP, its respective PB2-tip mutant failed to yield any detectable level of turbidity and spherical condensates (Fig. 6c, d and Supplementary Fig. 6g). A prolonged (3-day) incubation of CPB CP, but not the CPB CP PB2-tip mutant, yielded noticeably larger aggregates (Fig. 6c), hinting that the spherical condensates fuse with one another over time. Immunostaining with an anti-Plk4 antibody[4] confirmed that the PB2-tip mutation completely abrogated the formation of CPB CP condensates (Supplementary Fig. 6g). Thus, the PB2-tip region plays a key role in mediating phospho-PC3-induced Plk4 self-clustering.

To investigate whether CPB CP clustering exhibits any characteristics of phase separation, we induced CPB CP condensation in the presence of 5% 1,6-hexanediol, a weak hydrophobic interaction disruptor proposed as an agent to distinguish between liquid- and solid-like assemblies[41]. 1,6-Hexanediol has been shown to effectively disassemble several liquid-like assemblies, including nuclear pore complex[42,43] and stress granules[44–46]. We found that 1,6-hexanediol effectively obstructed the clustering of CPB CP in vitro (Fig. 6e). Treatment of EGFP-Plk4 CP-expressing cells with 1,6-hexanediol acutely diminished the level of Plk4 CP condensates on a timescale of seconds in vivo (Fig. 6f). In a related experiment, while fluorescent recovery after photobleaching (FRAP) showed that EGFP-Plk4 CP condensates recovered fluorescence within 2.5

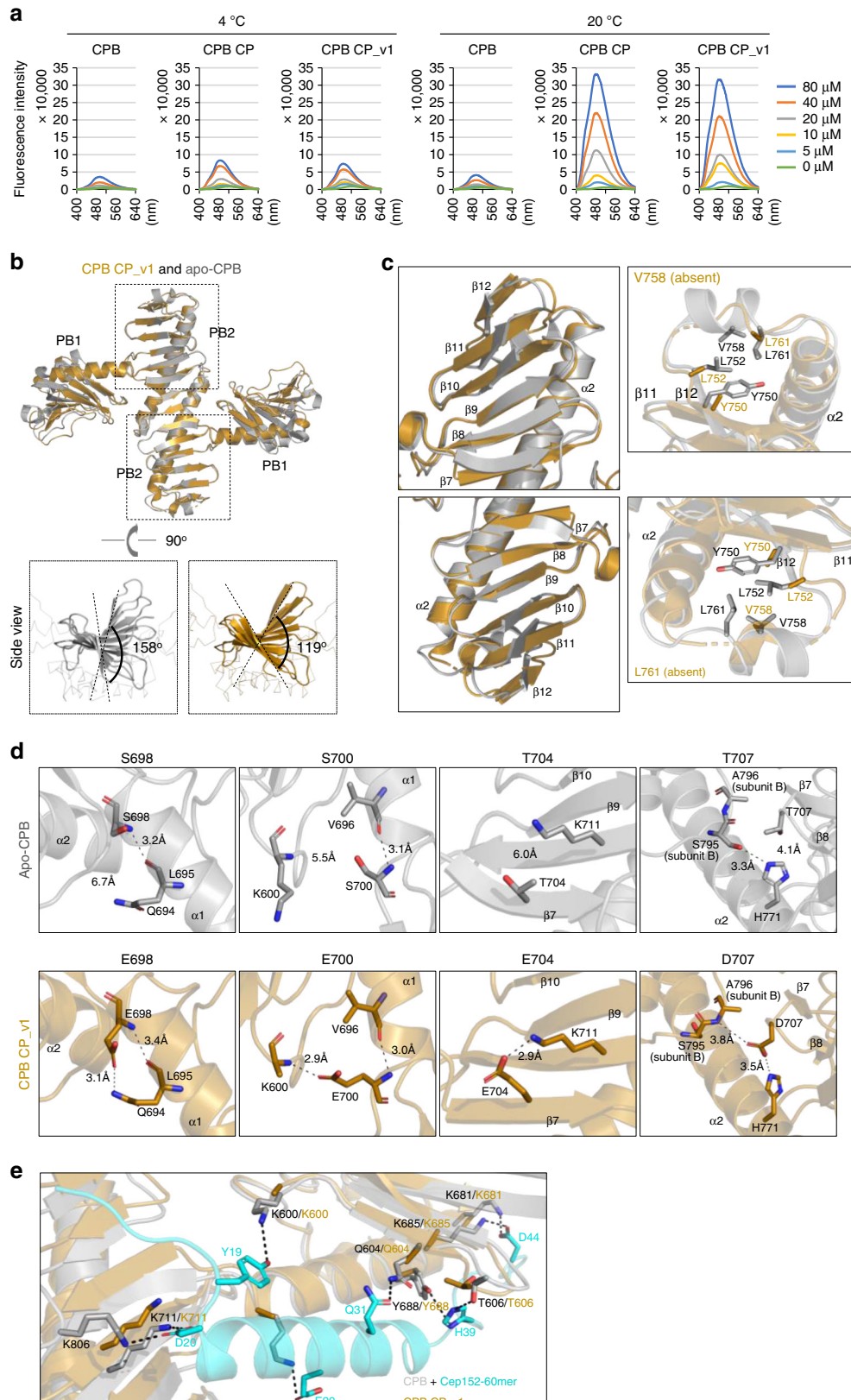

min, they failed to restore fluorescence in the presence of 1,6-hexanediol (Fig. 6g; Supplementary Video 3). Similar results were obtained with another protein-driven LLPS disruptor, α-lipoamide[47] (Supplementary Fig. 6h, i). Thus, although the pleiotropic nature of these small-molecule disruptors may not be ignored, these observations further support our notion that PC3-phosphorylated Plk4 can phase separate into biomolecular condensates. Notably, FRAP carried out with buffer-washed FITC–CPB CP condensates in vitro did not exhibit any detectable level of dynamic internal rearrangement (Fig. 6h), suggesting that these condensates may rapidly transition into an unexchangeable gel or solid-like state.

**Fig. 5** Altered surface hydrophobicity and conformational nature of condensation-proficient Plk4 CP and CP_v1. **a** Fluorescence spectra of ANS in a buffer containing the indicated concentrations of CPB, CPB CP, or CPB CP_v1 at 4 or 20 °C. **b** Overall structures of the CPB CP_v1 and apo-CPB (4N9J)[6] overlaid for comparison. Each subunit of a homodimeric CPB consists of PB1 and PB2 folds. Side views show the magnitude of β-sheet twisting along the PB2–PB2 axis (below). **c** Detailed structural comparison showing that the PB2-tip region of CPB CP_v1 is angularly distorted (left) and appears flexible with several disordered hydrophobic residues (right). Note that Y750, L752, V758, and L761 residues are either absent or devoid of their side chains. **d** Differences in the mode of interactions for PC3 WT and CP mutant residues. Important interactions (H-bond and/or salt bridge) are depicted as dashed lines (see text for details). **e** A hypothetical model illustrating that Cep152 60-mer residues (cyan) engaged in interactions with CPB residues (gray) fail to establish analogous interactions with CPB CP (brown), thus explaining why the CP mutations disrupted the Cep152-Plk4 interaction in Fig. 2d. Dotted lines indicate important interactions detected between Cep152 60-mer and CPB WT (4N7V). Source data are provided as a Source Data file for **a**

## Table 1 Data collection and refinement statistics

| | CPB CP_v1 | CPB CP |
|---|---|---|
| Data collection[a] | | |
| Space group | P32 | P32 |
| Cell dimensions | | |
| $a, b, c$ (Å) | 61.66, 61.66, 137.28 | 110.83, 110.83, 205.68 |
| $\alpha, \beta, \gamma$ (°) | 90.00, 90.00, 120.00 | 90.00, 90.00, 120.00 |
| Resolution (Å) | 50.0-2.64 (2.73-2.64) | 50.0-3.70 (3.83-3.70) |
| $R_{merge}$ | 0.098 (1.093) | 0.159 (0.978) |
| $I/\sigma I$ | 49.7 (2.8) | 23.75 (2.0) |
| Completeness (%) | 99.9 (100.0) | 100.0 (100.0) |
| Redundancy | 11.7 (11.6) | 11.0 (9.7) |
| Refinement | | |
| Resolution (Å) | 53.4-2.64 | 37.8-3.7 |
| No. of reflections | 16,254 | 29,745 |
| $R_{work}/R_{free}$ | 0.2562/0.3131 | 0.2709/0.3092 |
| No. of atoms | | |
| Protein | 2940 | 14,008 |
| Ligand/ion | 0 | 0 |
| Water | 0 | 0 |
| B factors | | |
| Mean | 82.37 | 167.13 |
| Max | 157.79 | 292.01 |
| Min | 30 | 75. 95 |
| R.m.s. deviations | | |
| Bond lengths (Å) | 0.015 | 0.005 |
| Bond angles (°) | 2.078 | 0.877 |
| Ramachandran | | |
| Favored | 91.00% | 97.84% |
| Allowed | 99.23% | 99.94% |
| Outliers | 0.77% | 0.06% |

[a]Statistics for the highest-resolution shell are shown within parentheses

To directly demonstrate whether Plk4-dependent CPB phosphorylation is sufficient to drive CPB self-clustering, we carried out in vitro kinase reaction using Sf9-purified GST-Plk4 (1–836) (i.e., GST-Plk4ΔPB3) and CPB, and examined the level of CPB clustering by conjugating the reaction product with FITC. Under these conditions, Plk4ΔPB3-phosphorylated CPB efficiently formed clusters, while unreacted CPB did not (Fig. 7a). In a second experiment, Plk4ΔPB3-reacted CPB was phosphorylated at the PC3 cluster (anti-pSSTT blot), and the resulting phospho-CPB was detected only in the pellet fraction (Fig. 7b, compare lane 2 with lane 3). These findings strongly suggest that Plk4-dependent PC3 phosphorylation leads to the formation of precipitable CPB condensates.

**PB2-tip mutations prevent Plk4 CP from inducing procentrioles**. Next, we investigated whether the PB2-tip-dependent clustering is important for Plk4-mediated centriole biogenesis. While Plk4 RNAi cells expressing Plk4 CP exhibited significantly increased centriole-loaded STIL, pS1108 epitope, and recruited

Sas6 signals as shown in Fig. 4d, mutations in the PB2 tip greatly diminished all three of these signals at centrosomes (Fig. 8a and Supplementary Fig. 7a, b). Notably, the PB2-tip mutant still exhibited a low level of these signals when compared with the catalytically inactive Plk4 KM mutant, suggesting that it possesses a remnant of Plk4's clustering property supporting its function.

To verify whether the PB2-tip-dependent clustering is critical for Plk4's ability to convert into a dot state, we carried out 3D-SIM analysis. As opposed to Plk4 CP-expressing cells showing dot-like or elongated Plk4 signals, the majority of the CP PB2-tip mutant displayed a ring-like localization (Fig. 8b, c and Supplementary Fig. 2g). However, unlike the catalytically inactive Plk4 KM incapable of disassociating from the Cep152 tether (Fig. 2d), the CP PB2-tip mutant was competent in not only dissociating from Cep152 but also binding to STIL (Supplementary Fig. 6e). These findings suggest that the CP PB2-tip mutant is defective specifically in its self-clustering ability and that STIL binding is not sufficient to drive Plk4's ring-to-dot conversion. As a consequence, the CP PB2-tip mutant exhibited dramatically diminished elongated Plk4 signals (Fig. 8d). A small fraction (~7%) of PB2-tip-mutant cells exhibiting elongated Plk4 signals could be attributable to a residual clustering ability that may still be present in the mutant. Taken together, the PB2-tip-mediated Plk4 clustering is required for Plk4 CP's ability to convert from a ring-state localization to a dot-state appearance and for Plk4-dependent centriole biogenesis.

## Discussion

Although dynamic relocalization of centriolar Plk4 from a ring-shaped pattern to a dot-like morphology has been reported[4–7], the molecular mechanism and biological significance of this process remain largely unknown. Our data provided here demonstrate that KD-dependent autophosphorylations at the CPB PC3 residues are sufficient to induce Plk4's ring-to-dot localization conversion through phospho-PC3-dependent self-clustering (Figs. 3 and 6). The CP PB2-tip mutant, defective in self-clustering but normal in STIL interaction (Fig. 6c, d and Supplementary Fig. 6e), failed to properly recruit STIL and Sas6 to the procentriole assembly site (Fig. 8a, b), which is to be expected if Plk4's ring-to-dot conversion is centrally required for centriole biogenesis. Thus, we propose that Plk4's stochastic autoactivation and phosphorylation at its PC3 motif induces local CPB clustering that serves as a trigger to break the symmetrical Plk4 localization around a centriole (Fig. 8e). Since Plk4 kinase activity promotes the Plk4–STIL interaction[5,20] (Fig. 2e and Supplementary Fig. 2j) and recruited STIL may activate Plk4 through a positive-feedback loop[22], the early symmetry-breaking process could be potentiated by an auto-amplification event(s), as proposed by extensive stochastic simulations[48]. Given that protein clustering can convert spatially distributed weak events into a focused all-or-none response, the robust clustering property of PC3-phosphorylated CPB will be crucial to efficiently phase-separate active Plk4 molecules to a dot-like condensate.

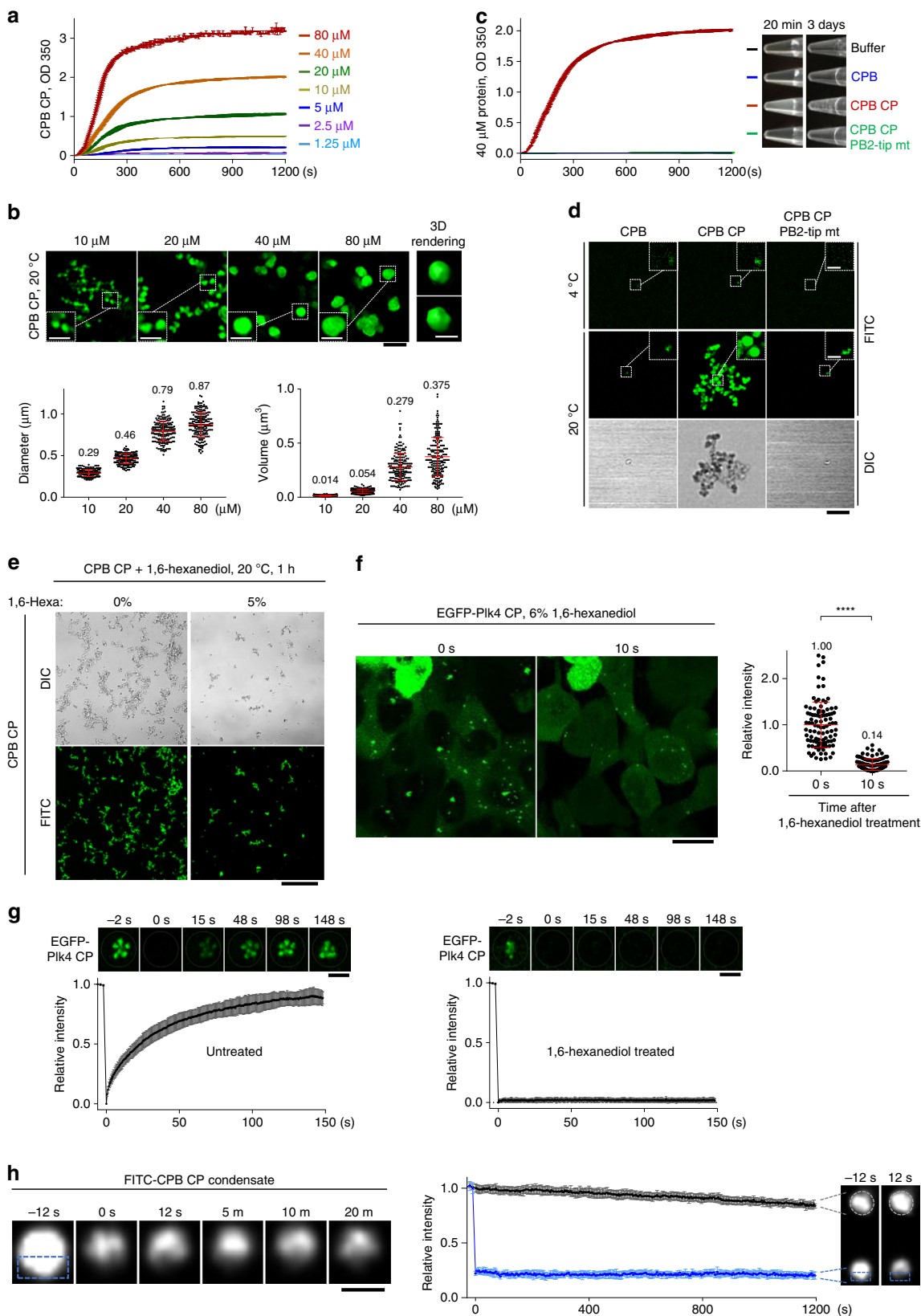

Data presented here demonstrate that Plk4 condensates are in an amorphous gel- or solid-like state that can be disrupted by weak hydrophobic interaction disruptors such as 1,6-hexanediol and α-lipoamide (Fig. 6e–g and Supplementary Fig. 6h, i). These observations suggest that Plk4 condensates behave like a phase-separating matrix-like body that can effectively trap its interacting partners from surrounding components for efficient downstream signaling (see below). Since Plk4 must reach a threshold level to

**Fig. 6** Autophosphorylated CPB is sufficient to generate spherical condensates in a concentration-dependent manner. **a** Condensation profiles of CPB CP at various concentrations as measured by $OD_{350\ nm}$ at 20 °C. Different concentrations of soluble CPB CP were prepared at 4 °C. Upon shifting the temperature to 20 °C, optical density was measured every 3 s. Each point represents the average of two replicates ± s.d. **b** 3D-SIM analysis showing the formation of spherical CPB condensates at various concentrations in vitro. Condensates formed at 20 °C were decorated with FITC for imaging (see Methods). Black bar, 2 μm. Dotted boxes, areas of enlargement (bar, 1 μm). A 3D surface-rendered spherical condensate (bar, 1 μm) is shown at two different angles. The diameter and volume of spherical condensates were quantified (mean ± s.d., n = 200/each concentration from three independent experiments). Numbers denote mean values. **c** Condensation profiles of the indicated proteins (40 μM) analyzed concurrently with **a** at 20 °C. Each point represents the average of two replicates ± s.d. Photographs showing the turbidity of each sample were taken after incubating 20 min or 3 days. **d** 3D-SIM images for the indicated proteins incubated at 4 °C (top) or 20 °C (bottom) for 1 h and decorated with FITC. Bar, 5 μm. Dotted boxes denote areas of enlargement (bar, 1 μm). DIC, Differential interference contrast. **e** DIC and 3D-SIM analyses for CPB CP condensates formed at 20 °C for 1 h in the presence of 5% 1,6-hexanediol and decorated with FITC. Bar, 20 μm. **f** Live cell imaging of U2OS cells stably expressing EGFP-Plk4 CP before (0 s) or after 6% 1,6-hexanediol treatment for 10 s. Bar, 20 μm. Quantification of relative fluorescence intensities is shown in mean ± s.d. (n = 101 for each timepoint from three independent experiments). **g** FRAP analysis for EGFP-Plk4 CP condensates in U2OS cells left untreated (left) or treated with 6% 1,6-hexanediol for 5 min (right). Images are from Supplementary Video 3. Bar, 2 μm. Relative signal intensities were quantified from 16 independent condensates for each group. Bars, s.d. **h** FRAP was carried out after photobleaching the bottom half (dotted box) of an FITC-decorated CPB CP condensate formed in vitro. A representative time-lapse series is shown (left). Bar, 0.5 μm. Relative signal intensities from 10 photobleached experimental hemispheres and their respective control condensates were quantified (right). Bars, s.d. Source data are provided as a Source Data file for **a–c** and **f–h**

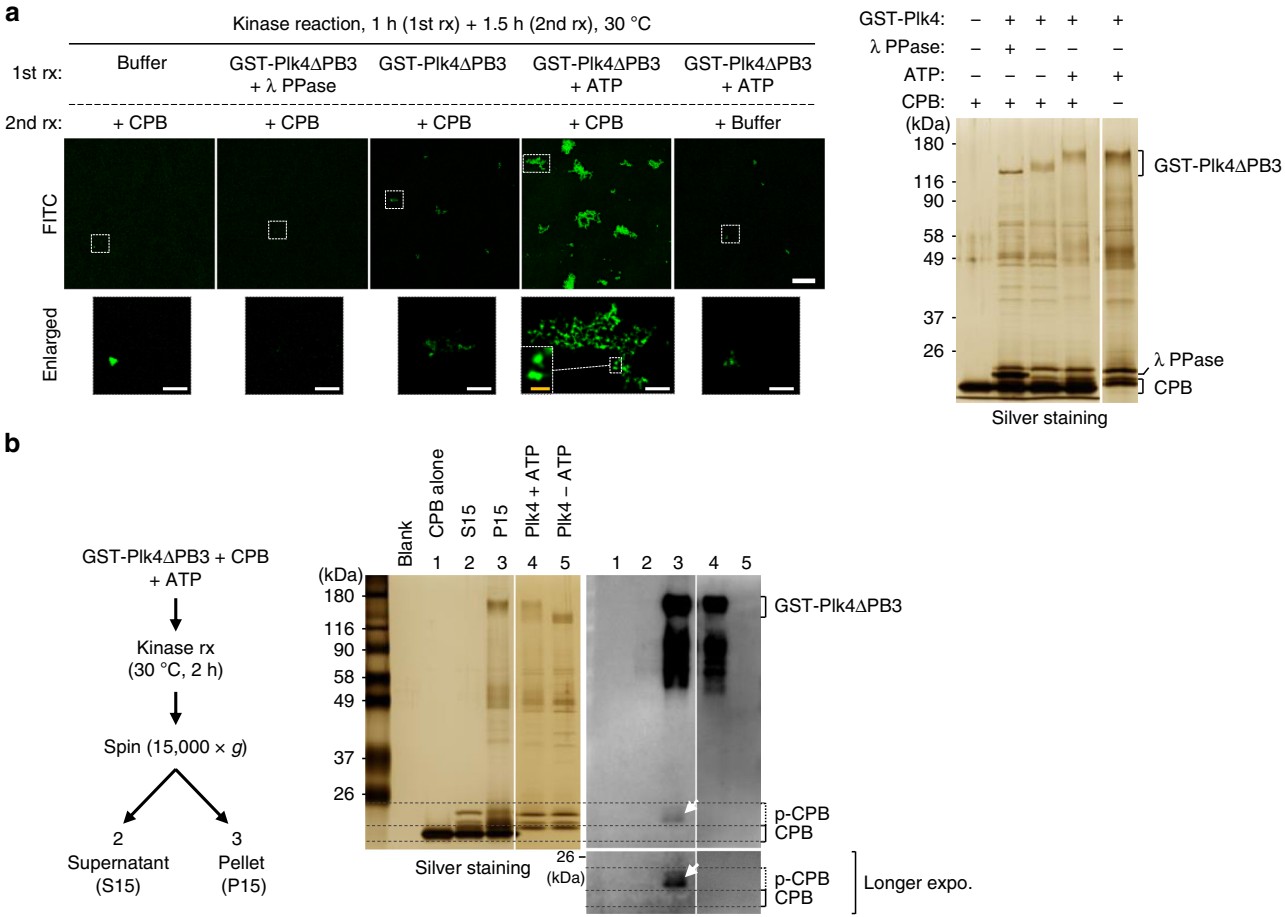

**Fig. 7** Phosphorylation of CPB by Plk4 induces clustering in vitro. **a** 3D-SIM images showing CPB condensates generated by GST-Plk4ΔPB3-dependent phosphorylation in vitro. Bar, 10 μm. Dotted boxes, areas of enlargement (bar, 2 μm). Yellow bar in the dotted box, 0.5 μm. Kinase reactions were carried out in two steps—Plk4ΔPB3 activation step (1st rx) and CPB phosphorylation step (2nd rx). Reaction products were decorated with FITC to visualize CPB condensates and imaged (left). The same reaction products were separated by 8% sodium dodecyl sulfate–polyacrylamide gel electrophoresis (SDS-PAGE) for silver staining (right). Note that FITC-decorated condensates contributed by GST-Plk4ΔPB3 (5th panel) are negligible. **b** Immunoblotting analysis showing Plk4ΔPB3-phosphorylated CPB generates pelletable condensates in vitro. Following a kinase reaction, a sample was fractionated into supernatant (S15) and pellet (P15), and then separated by 8% SDS-PAGE. Note that Plk4ΔPB3-reacted PC3-phosphorylated CPB (α-pSSTT panel) is present only in the P15 fraction (arrow)

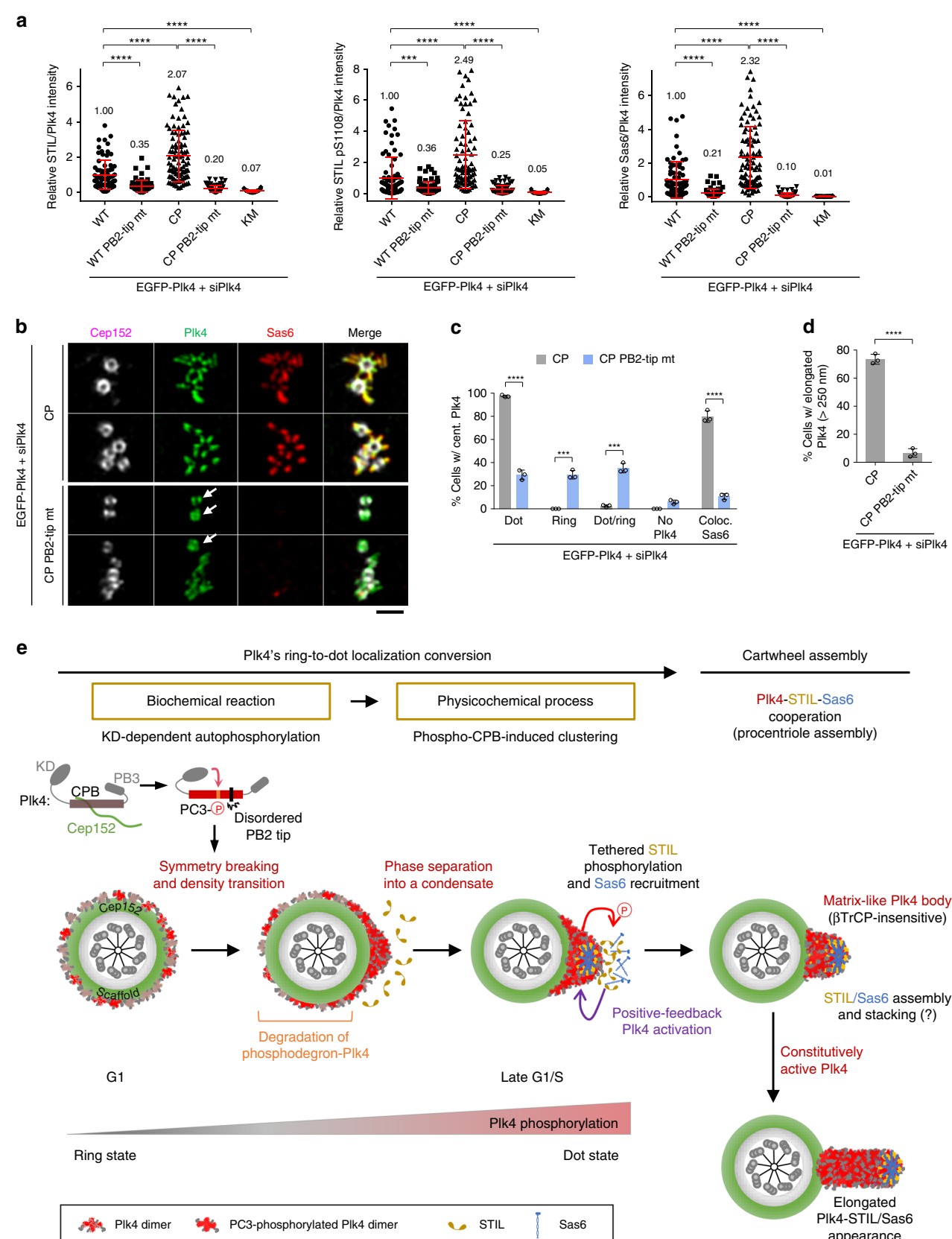

initiate centriole biogenesis[49] and because Sas6 can self-assemble into a cartwheel structure in a concentration-dependent manner[50], forming a Plk4 condensate would be important for recruiting a sufficient amount of the STIL–Sas6 complex that exceeds its threshold concentration for self-assembly. Since clustered Plk4 CP was largely insensitive to βTrCP-dependent proteasomal degradation (Fig. 4a, b), PC3-phosphorylated active Plk4 clusters could gradually accumulate over time, even under circumstances where recruited STIL can activate Plk4 and induce its proteasomal degradation[20,27]. These clusters may ultimately

**Fig. 8** Mutations at the PB2 tip disrupt Plk4's ability for ring-to-dot conversion, STIL and Sas6 recruitment, and procentriole formation. **a** Quantification of relative intensities of STIL, STIL pS1108, Sas6, and Plk4 fluorescence signals was carried out using confocal images acquired from immunostained U2OS cells stably expressing the indicated constructs under Plk4 RNAi conditions. The expression levels of Plk4 WT and mutants and their immunostained images are provided in Supplementary Fig. 7a, b. Relative fluorescence intensities (mean ± s.d.) are shown. $n = 75$ (WT), 75 (WT PB2-tip mt), 97 (CP), 81 (CP PB2-tip mt), and 78 (KM) for STIL/Plk4, $n = 80$ (WT), 75 (WT PB2-tip mt), 83 (CP), 77 (CP PB2-tip mt), and 75 (KM) for STIL pS1108/Plk4, and $n = 75$ (WT), 75 (WT PB2-tip mt), 97 (CP), 81 (CP PB2-tip mt), and 78 (KM) for Sas6/Plk4 with each sample pooled from three independent experiments. Numbers denote mean relative intensities. ***$P < 0.001$; ****$P < 0.0001$ (unpaired two-tailed $t$ test). **b** 3D-SIM analysis of immunostained U2OS cells stably expressing the indicated constructs under Plk4 RNAi conditions. Arrows denote the ring-state Plk4 CP PB2-tip mutant colocalized with Cep152. Bar, 1 μm. **c**, **d** Quantification of the samples in **b** is shown in mean ± s.d. ($n = 3$ independent experiments). ***$P < 0.001$; ****$P < 0.0001$ (unpaired two-tailed $t$ test). **e** Model illustrating the mechanism of how Plk4 breaks the symmetrical ring-state localization pattern and generates a dot-state assembling body for centriole biogenesis. In G1, as Cep152 is recruited to a region around a centriole, Plk4 interacts with Cep152 and localizes at the outskirts of the Cep152 scaffold (ring state)[6]. As predicted from extensive simulation analyses[48], a small fluctuation in the local concentration of Plk4 may allow its N-terminal catalytic activity to cross a critical point of phosphorylating the CPB PC3 motif, thus causing the phospho-CPB to cluster through the aggregative PB2-tip region (for simplicity, Plk4 is illustrated as a monomer) and generate a nanoscale condensate (dot state). The PC3 phosphorylation also promotes the Plk4–STIL interaction (Fig. 2). Since STIL activates Plk4[20,27], this interaction is expected to reinforce Plk4's ring-to-dot conversion in a positive-feedback fashion and enable Plk4 to rapidly coalesce into a centriole duplication-competent assembling body. The CP mutations at the PC3 motif enable Plk4 to bypass both Cep152-dependent pericentriolar localization and N-terminal KD-dependent CPB phosphorylation, and, therefore, to efficiently induce multiple dot-like or elongated procentrioles. Source data are provided as a Source Data file for **a**, **c**, **d**

coalesce into a matrix-like condensate that can foster an environment for centriole biogenesis (Fig. 8e). Subsequently recruited STIL and Sas6 may augment this process by further activating Plk4[20,27] and providing Plk4 clusters with additional protection from proteasomal degradation[5,27]. Thus, we propose that the dot-state Plk4 condensate serves as not only a stable reaction hub recruiting and efficiently phosphorylating PB3-tethered STIL but also an assembling body facilitating Plk4-mediated centriole biogenesis.

Our data shed light on the mechanism that determines how Plk4 forms a dot-like condensate prior to inducing centriole biogenesis. Recently, Gouveia et al.[24] showed that, while its molecular mechanism remains unknown, *Xenopus* Plk4 can form a spherical assembly in a catalytic activity-dependent manner. Unexpectedly, Yamamoto and Kitagawa[25] suggested that dephosphorylation of a flexible linker region (residues 280–305), previously characterized to be the phosphodegron motif for βTrCP[14], allows Plk4 to phase separate into condensates distributed around a centriole and induce centriole overduplication. However, since dephosphorylation of the βTrCP-binding motif would also stabilize Plk4, whether increased Plk4 stability contributed to the ring-like localization and centriole overduplication remains to be investigated.

We demonstrated that autophosphorylation at the PC3 motif of CPB is necessary and sufficient to drive the formation of dot-like Plk4 condensates in vivo (Fig. 2). At the biochemical level, a CPB CP mutant mimicking the PC3 phosphorylation exhibited dramatically increased accessible hydrophobic surfaces (Fig. 5a). This drastic change, which allows CPB CP to rapidly cluster and generate a spherical condensate (Fig. 6a, b), appeared to underlie the ability of Plk4 molecules to physically co-segregate and assume a dot-state localization from a pericentriolar ring-like pattern in vivo. Notably, although the condensation-prone CPB CP mutant exhibits some biomolecular condensate-like properties, such as its sensitivity to weak hydrophobic disruptors (Fig. 6e–g and Supplementary Fig. 6h, i), it rapidly formed spherical aggregates (Supplementary Fig. 6a) that failed to show any degree of internal rearrangement within a condensate (Fig. 6h). On the other hand, the full-length Plk4 CP mutant in both dot-like and elongated states exhibited a rapid turnover in U2OS cells (Supplementary Fig. 2d), suggesting either that its physicochemical properties could be somewhat different from those of the CPB CP mutant or that other elements may contribute to Plk4's dynamic nature *in vivo*. In a separate experiment, human Plk4 expressed in heterologous Sf9 cells, which would

minimize the effect of STIL or other Plk4-binding proteins, formed catalytic activity-dependent condensates (Supplementary Fig. 7c) that undergo fusion with other spherical assemblies and internal rearrangement of Plk4 molecules in young and less-dense condensates (Supplementary Fig. 7d, e). Young condensate-specific internal rearrangement was reported previously[51].

In this study, we demonstrated that Plk4 is a unique kinase whose KD-dependent autophosphorylation on its C-terminal CPB alters the physicochemical properties of the CPB, thereby causing it to undergo an acute density transition through phase-separating cluster formation. CPB PC3 phosphorylation and phospho-PC3-induced flexibility at the CPB PB2-tip appear to play a pivotal role in mediating this process (Fig. 8e). Whether other regions of CPB, such as the broader PB1 wing edge area, also reinforce this event remains to be further investigated. In addition, the mechanism by which the dot-state Plk4 disassembles and inactivates Plk4-dependent centriole duplication is another important area that needs further investigation. Remarkably, STIL and Sas6 are degraded after mitosis and reaccumulate in G1/S[52–54]. Thus, loss of STIL and/or other uncharacterized events, including Plk4 dephosphorylation, may weaken Plk4's clustering ability and allow Plk4 to revert to a ring state in early G1, as observed previously[4,5], before it becomes reactivated to initiate another round of centriole duplication.

## Methods

**Plasmid construction.** To generate various lentivirus-based EGFP-Plk4 constructs, a *Bam*HI–*Sal*I fragment containing the full-length WT (M4489), kinase-inactive K41M (KM) (M4491), PCR-mutagenized phosphocluster (pc) 1 (pc1) (T586A, S589A, T591A, S592A) (M4691), pc2 (S665A, S671A, S674A) (M4692), pc3 (S698A, S700A, T704A, T707A) (M4693), pc4 (T746A, S749A, T751A, S754A, S756A, S760A) (M4694), pc5 (T793A, S795A) (M4696), pc6 (S809A, T810A, S812A, S817A, S821A) (M4697), pc7 (T864A, 865A, 866A, S868A, T870A, S873A, 874A, 876A) (M5205), pc8 (S956A) (M4699), pc1–8 (all the combined mutations from pc1 to pc8) (M5208), or pc1–8 + PC3 [the pc1—8 mutations but with restored WT residues at the PC3 motif (S698, S700, T704, and T707)] (M5207) mutations was cloned into a pHR′.J-CMV-EGFP vector digested by *Bam*HI and *Sal*I. Similarly, construction of a full-length CP mutant (CP: S698E, S700E, T704E, and T707D) (M4899), a CP mutant on the kinase-inactive mutation (K41M) background (KM CP: K41M, S698E, S700E, T704E, and T707D) (M4903), a PB2-tip mutant (mt) (Y750A, L752A, V758A, and L761A) (M6562), a PB2-base mt (Y705F, K711A, and N717A) (M6564), a CP PB2-tip mt (S698E, S700E, T704E, T707D, Y750A, L752A, V758A, and L761A) (M6527), or a CP PB2-base mt (S698E, S700E, T704E, T707D, Y705F, K711A, and N717A) (M6534) were carried out by inserting a corresponding *Bam*HI–*Sal*I fragment into the pHR′.J-CMV-EGFP vector digested by *Bam*HI and *Sal*I. To generate the pKM7171 (EGFP-Plk4 CP_v1) construct expressing EGFP-Plk4 CP with the 32-residue insertion immediately after G808, a synthetic *Eco*RI–*Sal*I fragment containing both the insertion and Plk4 C terminus was used to replace the corresponding fragment in EGFP-Plk4 CP

(pKM4899). Generation of various lentiviral FLAG$_3$-Plk4 constructs was conducted by inserting a BamHI–SalI fragment containing the full-length WT (M4530), KM (M4532), pc3 (M4703), CP (M5070), KM CP (M5071), PB2-tip mt (M6565), PB2-base mt (M6567), CP PB2-tip mt (M6559), CP PB2-base mt (M6561), Plk4 CTD (581–970) (M4591), or Plk4 CTD CP (M5764) into a pHR′.J-CMV-FLAG$_3$ vector digested by the corresponding enzymes. All the Plk4 constructs containing siPlk4-insensitive silent mutations are marked with -sil. To generate constructs expressing Plk4 WT (pKM7162), Plk4 CP (pKM7164), and Plk4 CP PB2-tip mt (pKM7167) under their endogenous promoter control, a ClaI–EcoRI (internal site) fragment containing a CMV promoter-controlled Plk4 N terminus in pKM4489 was replaced with a ClaI–EcoRI fragment containing an endogenous promoter (nucleotide (nt) −360 to 0) and corresponding Plk4 WT or mutant N terminus.

Other lentiviral constructs designed to express Plk4 (1–808) (pKM6886), Plk4 CP (1–808) (pKM6887), or Plk4 CP (1–808)_v1 (pKM6889) were generated by inserting a BamHI–SalI (for pKM6886 and pKM6887) or BamHI–XhoI (for pKM6889) fragment into pHR′.J-CMV-FLAG$_3$ vector digested by BamHI and SalI. To construct pHR′.J-CMV-EGFP-STIL (718–868) (pKM5489), a BamHI–SalI fragment containing the respective STIL fragment was cloned into the pHR′.J-CMV-EGFP vector digested by the same enzymes.

For bacterial expression, a NdeI–XhoI fragment containing CPB (581–808) (pKM3686), CPB CP (581–808) (pKM6515), CBP CP PB2-tip mt (581–808) (pKM6892), or CPB CP_v1 (581–808) (pKM5850) was cloned into the pHis$_6$-MBP-TEV vector[55] digested by the corresponding enzymes. To generate the pET28a-His$_6$-CBP CP (581–808) construct (pKM5401), a NdeI–XhoI fragment containing the respective CBP CP fragment was inserted into the pET28a vector (Addgene) digested by the same enzymes.

For Sf9 cell expression, an AscI–SalI fragment containing EGFP-Plk4 was inserted into the pAcJ vector (pKM4565) derived from the pAcGHLT-A vector (BD Biosciences).

The pCI-neo-HA-STIL construct (pKM4483) was generated by inserting a PmeI–NotI fragment containing the full-length STIL into a pCI-neo-HA vector. The pEGFP-C1-Cep152 (1–217) construct (pKM3561) was reported previously[4]. pCDNA5/FRT/TO-Myc-GFP-STIL, -STIL (S1108A), -STIL (S1116A), and -STIL (S1108A S1116A) constructs[20] were kindly provided by Andrew J. Holland (Johns Hopkins, MD).

All the constructs used for this study are listed in Supplementary Table 1.

**Cell culture, transfection, and inhibitor treatment.** Human HEK293T, U2OS, and Sf9 cells were used for all the experiments carried out in this study. Cells were purchased from the American Type Culture Collection (ATCC). U2OS and HEK293T cells were cultured in McCoy's 5A (Thermo Fisher Scientific) and Dulbecco's modified Eagle's medium (Thermo Fisher Scientific) plus 1× GlutaMAX (Thermo Fisher Scientific) and 100 mM HEPES (Thermo Fisher Scientific), respectively, after supplementing them with 10% fetal bovine serum and 1× antibiotic–antimycotic (Thermo Fisher Scientific). Transfection was carried out using either PEI MAX (Polysciences) for protein expression or Lipofectamine RNAiMAX (Thermo Fisher Scientific) for siRNA-based knockdown of gene expression. For co-IP experiments with transfected HEK293T cells, all the constructs were cotransfected, unless otherwise indicated. Sf9 cells were cultured in Grace's insect medium (Thermo Fisher Scientific) supplemented with 10% fetal bovine serum. Baculovirus production and protein expression were carried out following standard laboratory procedures.

**Lentivirus production and generation of stable cell lines.** For the generation of lentiviruses expressing the gene of interest, HEK293T cells were cotransfected with pHR′-CMVΔR8.2Δvpr, pHR′-CMV-VSV-G (protein G of vesicular stomatitis virus), and a pHR′.J-CMV-SV-puro-based construct containing a target gene, using the calcium phosphate coprecipitation method previously described[56]. To generate stable cells, U2OS or HEK293T cells were infected for 1 day and then selected with 2 μg/mL of puromycin (Sigma-Aldrich). To minimize unintended overexpression, the amount of lentivirus showing ~95% survival rate after puromycin selection was used to infect the cells. The resulting U2OS stable cells were transfected with the appropriate siRNAs to knock down the respective endogenous proteins 1.5 days after lentivirus infection. To effectively deplete endogenous Plk4, a mixture of two siRNAs targeting Plk4 nt 2576–2596 and 3′ untranslated region was used. All the siRNAs used for this study are listed in Supplementary Table 2.

**Immunostaining.** Cells cultured on poly-L-lysine (Sigma-Aldrich)-coated No. 1.5 coverslips were fixed with 4% paraformaldehyde for 10 min and permeabilized with 0.5% Triton X-100 for 5 min at room temperature (RT). The resulting cells were placed in blocking buffer (5% bovine serum albumin [BSA] in phosphate-buffered saline [PBS]) for 1 h followed by washing, and labeled with the indicated primary antibodies (Supplementary Table 3) for 2 h at RT. Immunostaining with Alexa Fluor 594-conjugated anti-Plk4 pSSTT antibody was carried out after fixing the cells with 4% paraformaldehyde for 10 min and then treating them with methanol (−20 °C) for 2.5 min. All phosphoantibodies were used after preincubating them with their respective nonphospho-peptides (Supplementary Table 4) for 0.5 h at 4 °C. Samples were then washed five times with a washing buffer (PBS + 0.02% of Triton X-100) and labeled with an appropriate Alexa fluorophore-conjugated

secondary antibody (Supplementary Table 3) for 1 h at RT. After washing five times, samples were then mounted with Prolong Gold Antifade Reagent (Thermo Fisher Scientific) or SlowFade Gold Antifade Reagent (Thermo Fisher Scientific) for imaging.

**Confocal microscopy, 3D-SIM, and 3D surface rendering.** Confocal images were acquired using Zeiss LSM780 equipped with a plan-apochromat ×40 (NA 1.3) and ×63 (NA 1.4) oil-immersion lenses, 34-channel GaAsP spectral detector (Carl Zeiss Microscopy, LLC), and 12-bit, 0.5-μm z-steps. To quantify fluorescence signal intensities, images were acquired under the same settings and the images obtained after the maximum intensity projection of z-stacks were analyzed using the Zeiss ZEN v2.1 software (Carl Zeiss Microscopy, LLC).

3D-SIM was performed on a Zeiss LSM780 ELYRA S.1 microscope equipped with a ×63 plan-apochromat (NA 1.4) oil-immersion lens, 405/488/561/641 nm solid-state lasers, and excitation and emission filter sets for imaging x and y fluorescent labels. Acquired images were processed using SIM processing algorithm in the ZEN 2012 SP1 software.

3D surface rendering of 3D-SIM image stacks in Fig. 3c were generated using Imaris v.8.4.1 (Bitplane). Fluorescent beads (0.17 μm) were imaged to normalize the Z-stack scale of the objects.

**Live cell imaging.** For the results shown in Fig. 6f, U2OS cells stably expressing EGFP-Plk4 CP were cultured on a Lab-Tek II chambered coverglass with a No. 1.5 borosilicate glass bottom (Thermo Fisher Scientific), and then subjected to live cell imaging using a Zeiss LSM880 Airyscan microscope with a 32-channel GaAsP Airyscan detector, a plan-apochromat ×63 (NA 1.4) oil-immersion objective lens, and a Pecon heated stage incubator (temperature, humidity, and 5% CO$_2$ control) under the confocal imaging mode. The same area was imaged before and after treating the cells with 6% 1,6-hexanediol for 10 s under the same image acquisition settings. Relative signal intensities were quantified after maximum intensity projection using the ZEN v2.1 software.

**Visualizing Plk4 condensates with FITC isomer I.** To visualize CPB CP condensates generated in Fig. 6a with FITC isomer I (Sigma), 20 μM of soluble CPB CP in a buffer [20 mM Tris-HCl (pH 7.5), 500 mM NaCl, 5% (v/v) glycerol, and 0.5 mM TCEP (tris(2-carboxyl)phosphine)] was incubated in the Lab-Tek II chambered coverglass for 1 h at 20 °C and then conjugated with 40 μM of FITC for 20 min to decorate surface Lys residues. Decoration of CPB CP and GST-Plk4ΔPB3 condensates in Figs. 6e and 7a, respectively, was carried out similarly as above. The resulting FITC-conjugated aggregates were washed three times with PBS to remove unreacted FITC and then observed under 3D-SIM.

**Time-lapse microscopy.** For time-lapse microscopy in Supplementary Fig. 6a, an aliquot of FITC-conjugated CPB CP (20 μM) was loaded onto the chambered coverglass as described above and imaged every second up to 1000 s using the Zeiss LSM880 Airyscan microscope. The processed Airyscan images were analyzed using the Zen v2.1 software to quantify the mean fluorescence intensity in each individual object over time.

**Fluorescent recovery after photobleaching.** For FRAP imaging in Fig. 6h, FITC-conjugated CPB CP condensates prepared as above were subjected to time-lapse analysis using the Zeiss LSM880 Airyscan microscope. Photobleaching in the specific region of interest was performed using 100% acousto-optic tunable filter-modulated transmission power of the 488 nm laser, 20 iterations, and 2.05 μs pixel dwell time. Fluorescence recovery was monitored by collecting images with a 50 × 20-pixel image region of interest (ROI) (0.065 μm pixel size) every 4 s over a 1200-s time period. The Airyscan images were processed using the ZEN 2.3 SP1 image processing software and the processed images were analyzed using the FRAP module of the ZEN software to plot recovery curves.

To perform FRAP analysis in Supplementary Fig. 7e with EGFP-Plk4 condensates formed in Sf9 cells, cells were harvested 30 h post infection, sonicated for 10 s, and centrifuged at 1000 × g for 5 min to discard unbroken cells or large molecular weight debris. The resulting supernatant was further centrifuged at 15,000 × g for 10 min, washed with buffer [50 mM Tris-Cl 7.5, 10 mM MgCl$_2$, 2 mM EGTA, 2 mM dithiothreitol (DTT), PhosSTOP (Roche), and protease inhibitor cocktail (Roche)] twice, and then resuspended in the same buffer for analysis.

To compare the recovery ratio of EGFP-Plk4 CP condensates in vivo, U2OS cells stably expressing EGFP-Plk4 CP were cultured on the chambered coverglass as described above and then left untreated or treated with 6% of 1,6-hexanediol for 5 min (Fig. 6g) or 500 μM (0.01%) of α-lipoamide (6,8-dithiooctanoic amide; Santa Cruz Biotechnology) for 10 min (Supplementary Fig. 6i). Fluorescence recovery was monitored by collecting Airyscan images with a 67 × 53-pixel image ROI (0.132 μm pixel size), a condition that minimizes signal bleaching, every 500 ms over a 150-s time period.

**TEM and correlative light and electron microscopy.** For the thin-section TEM in Fig. 3 and Supplementary Fig. 3, U2OS cells stably expressing EGFP-Plk4 CP were

cultured on the Lab-Tek II-chambered coverglass described above. For immuno-TEM, samples were additionally stained with an anti-GFP antibody (Abcam, ab6556) and a goat anti-rabbit IgG-6 nm gold conjugate (Electron Microscopy Sciences). The resulting samples were fixed with 2% glutaraldehyde in a 0.1 M sodium cacodylate buffer (pH 7.4) for 1 h, treated with 1% OsO4 in the sodium cacodylate buffer for 1 h, and stained with 1% uranyl acetate in 0.1 M sodium acetate for 1 h. All the processes were conducted at RT. Subsequently, the sample was dehydrated and embedded in a Poly/Bed 812 embedding resin (Polysciences) and sectioned using an ultramicrotome (Leica EM UC7). The resulting 80-nm-thick sections were transferred onto a grid, and then stained with uranyl acetate and lead citrate. TEM images were obtained using a Hitachi H-7650 TEM.

For CLEM in Fig. 3 and Supplementary Fig. 3, U2OS cells stably expressing EGFP-Plk4 CP mutant were grown on a 35 mm No. 1.5 glass-bottom gridded dish (MatTek Corporation) and then fixed with 4% paraformaldehyde for 10 min. The cells with elongated centrioles were located and imaged using the Zeiss ELYRA 3D-SIM system (see above). After fixing the cells with 2% glutaraldehyde in the 0.1 M sodium cacodylate buffer (pH 7.4), the resulting samples were processed and embedded essentially as described above. The area containing the cells identified by 3D-SIM was sectioned and then prepared for TEM.

For the thin-section TEM in Supplementary Fig. 6c, purified CPB CP was incubated at 20 °C to trigger the formation of spherical condensates. The resulting sample was fixed with 2% glutaraldehyde in the sodium cacodylate buffer for 1 h at RT, and then processed for thin-section TEM.

**Negative-staining TEM.** To visualize the overall morphology of CPB and CPB CP by negative-staining TEM shown in Supplementary Fig. 5f, 10 μg/mL of purified CPB or CPB CP protein prepared in a buffer [20 mM Tris-HCl (pH 7.5), 500 mM NaCl, and 0.5 mM TCEP] was diluted and loaded onto a negatively charged carbon grid for 1 min, washed with filtered dH2O twice, stained with filtered 1% uranyl acetate (in H2O) for 1 min, and then dried after whisking away excess solution. The resulting sample was imaged using a TEM (Hitachi H-7650, Japan). To prevent the formation of CPB CP clusters by an elevated temperature, the entire negative-staining procedure was performed at 4 °C.

To obtain the negatively stained TEM image in Supplementary Fig. 6b, the same CPB CP sample (10 μg/mL) used for Supplementary Fig. 5f was loaded onto a negatively charged carbon grid at 20 °C to thermally induce condensation. Following 30 min incubation, the sample was processed for TEM as described above.

**Plk4 pSSTT, STIL pS1108, and Alexa Fluor-conjugated antibodies.** For the generation of rabbit polyclonal phosphoantibodies, synthetic phosphopeptides were used for immunization (Young In Frontier, South Korea). Phosphoantibodies were affinity-purified using a corresponding phospho-peptide immobilized to SulfoLink resin (Thermo Fisher Scientific). Alexa Fluor 594-conjugated anti-Plk4 pSSTT antibody was generated using the Alexa Fluor 594 protein labeling kit (Molecular Probes). The list of phosphopeptides and their respective nonphospho-peptides used in this study is provided in Supplementary Table 4.

An anti-Cep152 (491–810) antibody[4] was conjugated with Alexa Fluor 647 by using the Alexa Fluor 647 protein labeling kit (Molecular Probes).

**IP, λ PPase treatment, and immunoblotting.** IP was carried out essentially as described previously[57] in TBSN buffer [20 mM Tris-Cl (pH 8.0), 150 mM NaCl, 0.5% NP-40, 5 mM EGTA, 2 mM MgCl2, 1.5 mM EDTA, 2 mM DTT, 20 mM p-nitrophenyl phosphate (PNPP), and protease inhibitor cocktail (Roche)]. Where indicated, immunoprecipitated proteins were treated with λ phosphatase for 1 h and then separated by sodium dodecyl sulfate-polyacrylamide gel electrophoresis (SDS-PAGE). To treat the total lysates with λ phosphatase, cells were lysed in a λ phosphatase buffer [50 mM HEPES (pH 7.5), 150 mM NaCl, 0.4% NP-40, 2 mM DTT, 1 mM MnCl2, and protease inhibitor cocktail (Roche)], incubated with λ phosphatase (New England Biolab) for 1 h at 30 °C, and then subjected to immunoblotting.

Immunoblotting analysis was performed according to standard procedures using an enhanced chemiluminescence detection system (Thermo Fisher Scientific). To detect specific phosphoepitopes (i.e., Plk4 pSSTT and STIL pS1108), immunoblotting was carried out in the presence of their respective nonphospho-peptide. Immunoblot signal intensities were quantified using the Image J (Fiji) or Image Lab 5.2.1 (Bio-Rad) program. The specific protein intensity was determined after subtracting non-specific background signals. All the antibodies used for this study are listed in Supplementary Table 3. Detailed information will be provided upon request.

**MS analysis.** Asynchronously growing HEK293T cells transfected with EGFP-Plk4 were subjected to IP with an anti-GFP antibody in the presence of PhosSTOP (Sigma-Aldrich). The resulting immunoprecipitates were separated by 7.5% SDS-PAGE and stained with Coomassie Brilliant Blue. Hyperphosphorylated slow-migrating Plk4 species was clearly detectable, as shown Supplementary Fig. 1b. Phosphorylated Plk4 excised from the gel was in-gel digested with trypsin P, chymotrypsin, LysC, ArgC, GluC, or AspN to extract the peptides[58]. After desalting

by C18 ZipTip (Millipore), the extracted samples were subjected to liquid chromatography-MS (LC-MS) analysis.

For LC-MS analysis and data processing, each sample (6 μL) was loaded on an Easy nLC II nano-capillary HPLC system (Thermo Fisher Scientific) with a C18 Nano-Trap Column, (Thermo Fisher Scientific) and a C18 Nano analytical column (15 cm, nanoViper, Thermo Fisher Scientific) connected with a stainless-steel emitter, coupled online with a Dual-Pressure Linear Trap MS spectrometer (LTQ Velos, Thermo Fisher Scientific) for MS analysis. Peptides were eluted using a linear gradient of 2% mobile phase B (acetonitrile with 0.1% formic acid) to 42% mobile phase B within 45 min at a constant flow rate of 200 nL/min. The 15 most intense molecular ions in the MS scan were sequentially selected for collision-induced dissociation using a normalized collision energy of 35%. The MS spectra were acquired at the MS range of m/z 300–2000. Nanospray Flex™ Ion Sources (Thermo Fisher Scientific) capillary voltage and temperature were set at 1.7 kV and 300 °C, respectively. The dynamic exclusion function on the MS spectrometer was enabled during the MS2 data acquisition with a 30-s window. The MS data were searched against Plk4 protein sequence utilizing BioWorks 3.3.1 Server Edition. Up to two missed tryptic cleavage sites was allowed. The oxidation of methionine, carbamidomethylation of cysteine, and phosphorylation on serine, threonine, and tyrosine were included as a dynamic modification. The precursor ion tolerance was set at 1.4 Da and the fragment ion tolerance was set at 0.6 Da. The data have been deposited in MassIVE (Accession #: MSV000084261).

The BioWorks software reported multiple phosphorylation site assignments for the peptides within the pS698 through pT707 region and manual sequencing of the spectra was required. As shown in Supplementary Fig. 2c, equivalent evidence existed to support the presence of two phosphopeptides differing in only the localization of the third phosphorylation site.

The ArgC-digested peptide search identified the Plk4 peptide SKSPKITYFTR and indicated a phosphorylation on pS698, pS700, and pT704. To further evaluate the MS2 spectra, the +2 charged parent ion 784.53 was extracted from the chromatograph revealing a wide peak of 1.2 min allowing ample time for other form of the peptides to be eluded during the same window. To evaluate if another form of the peptide was present with identical parent mass, the spectra was annotated manually. We specifically searched for a triply phosphorylated form, looking specifically for the y2 = 356.13, y3 = 503.20, and y4 = 666.27 that would support the presence of the pS698, pS700, and pT707 form of the peptide.

**Kinase reaction and silver staining.** In vitro kinase reaction was carried out essentially as previously described[57]. For two-step kinase reactions in Fig. 7a, reactions were carried out with 27 nM of Sf9-purified GST-Plk4 (1–836) (Sigma-Aldrich) under the conditions indicated (1st reaction, 30 °C, 1 h) and then subjected to subsequent reactions with 3 μM of CPB or control buffer (2nd reaction, 30 °C, 1.5 h). A part of reaction products was mounted on a coverslip, conjugated with FITC, and subjected to 3D-SIM. Remaining reaction samples were separated by 8% SDS-PAGE and stained with silver.

For kinase reactions in Fig. 7b, 27 nM of GST-Plk4 (1–836), 3 μM of CPB, and/or 100 μM ATP were used as indicated. To assess whether phosphorylation of CPB by GST-Plk4 is sufficient to induce precipitable condensates, the reaction carried out in the presence of both CPB and ATP was centrifuged at 15,000 × g for 10 min to fractionate into supernatant (S15) and pellet (P15). The resulting samples were separated by 8% SDS-PAGE (P15 was loaded three times more than S15) for silver staining and immunoblotting analysis.

**Size-exclusion chromatography.** To determine their dimeric state in solution, an equal amount of purified apo-CPB, CPB CP, or CPB CP_v1 (see the purification method below) was loaded onto Superdex 75 Increase 10/300 GL (GE Healthcare) connected to ÄKTA Pure 25 L1 (GE Healthcare). Samples were eluted in a buffer [20 mM Tris-HCl (pH 7.5), 500 mM NaCl, 5% (v/v) glycerol, and 0.5 mM TCEP] at the flow rate of 0.5 mL/min for 60 min. The data were collected and analyzed by Unicorn 7 software (GE Healthcare). An equal amount of BSA (Sigma-Aldrich) was analyzed under the same conditions for comparison.

**Protein expression and purification.** CPB CP (pKM6515) and CPB CP_v1 (pKM5850) were expressed in an Escherichia coli Rosetta strain (Novagen). Cells were cultured in Luria Broth medium containing 25 μg/mL of chloramphenicol and 50 μg/mL of kanamycin at 37 °C until their OD600 reached to 0.6–0.8. The cells were then treated with 0.3 mM isopropyl β-D-1-thiogalactopyranoside and continuously cultured at 16 °C overnight before harvesting them by centrifugation at 4000 r.p.m. at 4 °C. The resulting cells were lysed in an ice-cold binding buffer [20 mM Tris-HCl (pH 7.5), 500 mM NaCl, 10% (v/v) glycerol, and 0.5 mM TCEP] by ultrasonication. His6-MBP-fused CPB CP was captured using a 5-mL HisTrap HP column (GE Healthcare) pre-equilibrated with the binding buffer and eluted with the binding buffer with a step gradient of 25, 50, or 500 mM imidazole. The sample was further subjected to size-exclusion chromatography (SEC) with HiLoad 16/600 Superdex 200 pg column (GE Healthcare) in a buffer [20 mM Tris-HCl (pH 7.5), 500 mM NaCl, 5% (v/v) glycerol, and 0.5 mM TCEP]. The collected sample was digested with a recombinant tobacco etch virus protease (TEV) on ice, overnight, to cleave off the His6-MBP tag, loaded onto 5-mL HisTrap HP and 5-mL MBPTrap HP columns (GE Healthcare) to remove His6-MBP and His6-TEV, and

then subjected to another SEC with HiLoad 16/600 Superdex 200 pg column to eliminate contaminated proteins. The purification procedure for CPB CP_v1 is essentially the same as for CPB CP. All the purification steps were performed either at 4 °C or on ice. Purified CPB CP and CPB CP_v1 exhibited >95% purity based on SDS-PAGE analysis.

To purify His$_6$-CPB CP (pKM5401), Rosetta cells expressing His$_6$-CPB CP were cultured and lysed essentially as described above. His$_6$-CPB CP was eluted from a 5-mL HisTrap HP column after binding and loaded onto SEC with HiLoad 16/600 Superdex 200 pg column in a buffer [20 mM Tris-HCl (pH 7.5), 500 mM NaCl, 5% (v/v) glycerol and 0.5 mM TCEP]. Based on SDS-PAGE analysis, the purity of the protein was estimated to be >95%.

**Crystallization and data collection.** For crystallization, CPB CP_v1 (pKM5850) was concentrated to 17 mg/mL in a buffer containing 20 mM Tris-HCl, pH 7.5, 500 mM NaCl, 5% (v/v) glycerol, and 0.5 mM TCEP. Two microliters of protein was mixed at a ratio of 1:1 with a reservoir solution containing 4.3 M ammonium acetate and 0.1 M Bis-Tris propane (pH 8.0) and crystals were grown at 20 °C for 3 days with a sitting-drop vapor-diffusion method.

In order to crystalize His$_6$-CPB CP (pKM5401), the protein was concentrated to 3.5 mg/mL in the same buffer as above. Three microliters of protein were mixed at a ratio of 1:1 with a reservoir solution containing 22.5% (w/v) PEG2000, 3% (w/v) dextran sulfate sodium salt, 0.1 M bicine (pH 8.5), 24 μM nonaethylene glycol monododecyl ether (C12E9), and 8% (v/v) glycerol, and crystals were grown at 8 °C for 2 days with a sitting-drop vapor-diffusion method. Under these conditions, we were able to obtain CPB CP crystals by minimizing its coagulating activity.

Crystals were flash frozen in liquid nitrogen after soaking them in the reservoir solution supplemented with an additional 20% (v/v) glycerol as a cryoprotectant. Diffraction data sets were collected at 100 K on the 22-ID beam line of the Advanced Photon Source, Argonne, Illinois.

**Structure determination.** Diffraction data sets were indexed, integrated, and scaled with HKL-2000[59]. The crystal structure of CPB CP_v1 was solved by molecular replacement method using phaser from the CCP4 software suite[60,61]. The crystal structure of apo-CPB (PDB code: 4N9J) was used as an initial model. Further refinement was carried out with refmac5 in CCP4 software suite[62] and phenix.refine in Phenix suite[63]. Data collection and refinement statistics are summarized in Table 1.

The structure of the final CPB CP_v1 model was verified using Phenix suite. Water molecules were not identified likely because of a high Wilson $B$-factor. The Ramachandran plots showed 91.00%, 99.23%, and 0.77% of the residues in the favored, allowed, and outlier regions, respectively. All structural figures were rendered using the PyMOL 2.2.0 software (PyMOL Molecular Graphics System, Schrödinger, LLC).

The structure of His$_6$-CPB CP was determined using a combination of RosettaCM[64] and Rosetta/Phenix crystallographic refinement[65]. An initial model of a dimer of apo-CPB (PDB code: 4N9J) was used as a molecular replacement model in Phaser. This search yielded a good solution (TFZ = 16.4) with four copies of the dimer in the asymmetric unit. To rebuild and refine the His$_6$-CPB CP model and to corroborate the effect of the CP mutation on the CPB structure, the apo-CPB structure, but not the CPB CP_v1 structure, was used as the initial model to eliminate a potential structural bias towards the CP mutant. Due to the low resolution of the dataset, several tools in Rosetta were used. First, a 2mFo-DFc density map was calculated from the initial MR hit, and RosettaCM was used to rebuild and refine each of the four dimer copies into density. The best model—corresponding to chains G and H in the final refined model—was propagated to all four dimer copies, and the full complex was refined using Rosetta/Phenix reciprocal space refinement.

A final cycle of $B$-factor refinement without hydrogen was carried out in Phenix, and the final structure of His$_6$-CPB CP was verified using Phenix suite. The Ramachandran plots showed 97.84%, 99.94%, and 0.06% of the residues in the favored, allowed, and outlier regions, respectively.

**ANS fluorescence measurement.** To estimate an accessible hydrophobic surface available on a protein, various concentrations of purified CPB, CPB CP, or CPB CP_v1 were incubated with 200 μM of ANS at 4 °C or 20 °C for 2 h. The resulting samples were transferred into 384-well microplate (Aurora) and the intensity of ANS fluorescence was monitored using a Perkin-Elmer EnSpire Multimode Plate reader. Excitation was carried out at 365 nm and emission was scanned between 400 and 650 nm, with 1 nm bandpass used at both excitation and emission slits.

**Optical density analysis.** To investigate the clustering kinetics of purified CPB, CPB CP, or the CPB CP PB2-tip mutant, optical density was measured every 3 s for a total of 1200 s at 350 nm at 20 °C using a Beckman Coulter DU 800 spectrophotometer.

**Statistics and reproducibility.** All values are provided as mean of $n \pm$ s.d. The statistical details including the definitions and value of $n$ (e.g., number of experimental replicates, cells, centrioles, condensates, etc.) and standard deviations are provided in the figures, corresponding figure Legends, and results. Statistical significance was assessed using an unpaired two-tailed $t$ test in GraphPad Prism (*$P <$ 0.05; **$P < 0.01$; ***$P < 0.001$; ****$P < 0.0001$).

**Reporting summary.** Further information on research design is available in the Nature Research Reporting Summary linked to this article.

## Data availability

Mass spectrometry data have been deposited in MassIVE (https://massive.ucsd.edu/ProteoSAFe/static/massive.jsp) (Accession #: MSV000084261). The coordinates and structure factors of CPB CP_v1 (6N45) [https://doi.org/10.2210/pdb6N45/pdb] and His$_6$-CPB CP (6N46) [https://doi.org/10.2210/pdb6N46/pdb] have been deposited in the Protein Data Bank. All relevant raw data are available from the authors. Numerical source data underlying Figs. 1c, d, 2a, b, c, 3c, 4a, b, d, e, 5a, 6a, b, c, f, g, h, 8a, c, d and Supplementary Figs. 1a, f, 2a, b, d, f, g, 4a, d, e, 5c, h, 6a, i, 7c, e are provided as a Source Data file. Source data (raw gels and blots) for Figs. 1d, 2d–f, 4a, 7a, b and Supplementary Figs. 1b, e–g, 2a, e, h–k, 4b–d, 5b, c, 6d, e, 7a, c are provided in Supplementary Fig. 8.

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

## Acknowledgements

We are grateful to Lingjun Meng for initial crystallization of CPB CP_v1, Michael Kruhlak and Langston Lim for assisting with time-lapse microscopy, Kunio Nagashima and Louis (Chip) Dye for providing technical service with electron microscopy samples, Ming Zhou and Benjamin C. Orsburn for mass spec data search and analysis, Di Xia for structural analysis, Yanling Liu for generating 3D surface-rendered movies, Richard J. Wheeler (Oxford University, UK) for sharing unpublished data about α-lipoamide, Karen Oegema for providing centrinone (LCR-263), and Raymond Erikson for critical reading of the manuscript. This research was supported by the Intramural Research Program of the National Institutes of Health, National Cancer Institute (K.S.L.) and an NST grant CAP-16-03-KRIBB of South Korea (J.K.B.).

## Author contributions

J.-E.P. and K.S.L. designed all the experiments, and J.-E.P. performed all the experiments except crystal X-ray diffraction data analysis and MS. J.-E.P. generated and optimized CPB CP crystals and obtained diffraction data, while F.D. solved the 3.7 Å CPB CP structure. L.Z. optimized CPB CP_v1 crystals, solved the 2.64 Å structure, and carried out comparative structural analyses among different CPB variants. T.A. carried out MS for immunoprecipitated EGFP-Plk4 and analyzed the data. J.K.B synthesized phosphopeptides and generated antisera. K.S.L., J-E.P., L.Z., F.D., and T.A. wrote the manuscript.

## Competing interests

The authors declare no competing interests.
