## [Peer Review File · Nature Communications]

REVIEWERS' COMMENTS:

Reviewer #1 (Remarks to the Author):

–The current manuscript by Park et al., has been improved and some of my major concerns have now been addressed. I still have a small number of concerns that I think the authors should try to address before publication in Nature Comms.

- The authors need to show in their own hands that STIL preferentially binds to kinase activate and not inactive PLK4. This is a critical aspect of the model and the experiments that were performed all used PPases in the IP procedure that nullifies the difference between the wildtype and kinase-dead PLK4. I realize that a previous study made the claim that STIL only binds to kinase active PLK4, but this observation has been contradicted by others in the literature, so it is critical to know what the authors find in their hands.

- The PC3 mutant shows increased abundance and inability to promote SAS6 recruitment. These phenotypes are consistent with the interpretation that PC3 EGFP-Plk4 is kinase inactive/compromised. The authors should test whether or not this is not the case?

- Figure 1d. Why does CEP152 form punctate cytoplasmic structures that don't overlap with PLK4 or centrosomes in the cells overexpressing PC3 EGFP-Plk4?

- Figure S2e. The expression of the PLK4 CP mutant is higher than that of the wildtype PLK4 construct. I, therefore, feel the authors should acknowledge that the differences they see could be, at least in part, caused by the difference in expression levels for these two constructs.

- The authors state that PC3-phosphorylated PLK4 can drive procentriole assembly independently of CEP152. However, the elongated SAS6 assemblies formed with this mutant are not procentrioles. This claim should, therefore, be toned down. Furthermore, on page 13 the authors state “the aggregative Plk4 CP mutant’s ability to potently induce procentrioles”, while on page 14 they state, “As expected, pSSTT and pS1108 epitopes and Sas6 signals were also manifest along the entire length of Plk4 CP-induced elongated procentrioles”. Again, I don’t think the authors have convincingly shown that the PLK4 CP mutant produces bone fide procentrioles at a higher frequency than wildtype PLK4. The assemblies generated by the PLK4 CP mutant are nonphysiological and should authors should be more cautious in their description.

- I felt the experiment in Figure S7c was very important as it shows full-length kinase active PLK4 forms condensates in insect cells. This is critical because most of the experiments in the paper are performed in the background of the CP mutations, which may not faithfully mimic phosphorylation of the PLK4 CPB. It would be great if the authors can show that the PB2-tip mutations disrupt the PLK4 condensates formed in the insect cells. I know these experiments are time-consuming and I wouldn't insist on them for publication. However, since the PB2-tip mutations are loss-of-function mutants, it is not surprising they can overcome the gain-of-function CP phenotypes. Showing the PB2-tip mutations disrupt condensate formation in the context of full-length wildtype PLK4 would be important to interpret how the loss-of-function PB2-tip mutations affect PLK4 function.

Responses to the reviewers' comments:

REVIEWERS' COMMENTS:

Reviewer #1 (Remarks to the Author):

–The current manuscript by Park et al., has been improved and some of my major concerns have now been addressed. I still have a small number of concerns that I think the authors should try to address before publication in Nature Comms.

- The authors need to show in their own hands that STIL preferentially binds to kinase active and not inactive PLK4. This is a critical aspect of the model and the experiments that were performed all used PPases in the IP procedure that nullifies the difference between the wildtype and kinase-dead PLK4. I realize that a previous study made the claim that STIL only binds to kinase active PLK4, but this observation has been contradicted by others in the literature, so it is critical to know what the authors find in their hands.

(This is a new question came up when reviewing the revised version). Perhaps, we did not describe our experimental procedure clearly. The immunoprecipitations were carried out without treating the lysates with λ PPase, so that the difference between the WT and its catalytically inactive form can be seen. After immunoprecipitation, the resulting immunoprecipitates were treated with λ PPase to convert all the phosphorylated and slow-migrating forms into a single fast-migrating form for accurately determining the amounts of co-precipitates. To avoid this confusion, we now stated that “IP’ed samples were then treated with λ phosphatase, where indicated, to convert phosphorylated----” in Fig. 2 and Supplementary Fig. 2h legends.

- The PC3 mutant shows increased abundance and inability to promote SAS6 recruitment. These phenotypes are consistent with the interpretation that PC3 EGFP-Plk4 is kinase inactive/compromised. The authors should test whether or not this is not the case?

(This is a new question came up when reviewing the revised version). The Reviewer #1 wonders whether the pc3 mutant may have lost its catalytic activity, because its abundance is somewhat similar to that of the catalytically inactive Plk4 KM mutant. However, the potent procentriole assembly observed with the gain-of-function, condensation-prone CP mutant (bearing the phospho-mimicking Asp or Glu residues at the PC3 site) suggests that the pc3 mutant is an underphosphorylated and less-active form that can be further activated by the CP mutations. Consistent with this view, unlike the catalytically inactive Plk4 KM mutant, the pc3 mutant exhibited a significant level of ring-to-dot conversion (Fig. 1c) and Sas6 recruitment to centrosomes (Fig. 1d and Supplementary Fig. 1f). In addition, to aid readers better appreciate this point, we provide an extra data that demonstrates that the Plk4 pc3 mutant transphosphorylates a catalytically inactive Plk4 KM substrate (slow-migrating Flag-Plk4; red arrows) and induces its degradation (note that phosphorylation induces proteasomal degradation) almost as efficiently as WT (Supplementary Fig. 1g; compare lane 3 with lane 7). Under the same conditions, the catalytically inactive Plk4 KM mutant did not show any of these activities (lane 5). This is described in line 108 in the main text and Supplementary Fig. 1g legend. However, we agree with the reviewer that the PC3 sites are some of the major phosphorylation sites whose mutations result in hypophosphorylated fast-migrating Plk4 forms (Supplementary Fig. 1e). Thus, to accommodate the reviewer’s any remaining concerns, we toned down and stated in the Supplementary figure legend that “pc3 mutations may compromise β TrCP-dependent proteasomal degradation of Plk4.”

- Figure 1d. Why does CEP152 form punctate cytoplasmic structures that don't overlap with PLK4 or centrosomes in the cells overexpressing PC3 EGFP-Plk4?

(This is a new question came up when reviewing the revised version). There might be some variations from one cell to another. However, as one can see in Fig. 1d, the 3rd panel from top, four other cells expressing the Plk4 pc3 mutant at a lower level (i.e., not overexpressed) clearly show that the pc3 mutant colocalizes with Cep152 at centrosomes.

- Figure S2e. The expression of the PLK4 CP mutant is higher than that of the wildtype PLK4 construct. I, therefore, feel the authors should acknowledge that the differences they see could be, at least in part, caused by the difference in expression levels for these two constructs.

The Plk4 CP mutant expressed under its endogenous promoter exhibited a twofold-increased expression level when compared with its respective WT (Supplementary Fig. 2e). This is likely because the CP mutant becomes somewhat insensitive to β TrCP-dependent proteasomal degradation, as shown in Fig. 4a. The Reviewer #1 concerns whether this increase may have contributed to the ability of the CP mutant to potently induce elongated procentrioles. Notably, the CP PB2-tip mutant bearing mutations at the PB2-tip was literally defective in procentriole formation (Supplementary Fig. 2f,g), even though its expression level was almost equal to that of its respective CP mutant (Supplementary Fig. 2e). Furthermore, comparative analyses of the signal intensity of recruited Sas6 per centrosome-localized Plk4 intensity showed that the Plk4 CP mutant was 2.3-fold more efficient than Plk4 WT (Fig. 8a). Again, its respective Plk4 CP PB2-tip mutant was largely defective in recruiting Sas6. Therefore, although we cannot eliminate the possibility that the increased level of Plk4 CP contributed to Plk4 CP-induced procentriole assembly, these data strongly suggest that Plk4 CP has a constitutively active gain-of-function activity in recruiting STIL and Sas6 and driving centriole biogenesis. However, since we cannot eliminate the contribution coming from the increased level of the CP mutant, we incorporated this potential concern in line 149 “Although Plk4 CP’s twofold-increased expression level (Supplementary Fig. 2e) may have contributed to its capacity to induce centriole biogenesis---.”

- The authors state that PC3-phosphorylated PLK4 can drive procentriole assembly independently of CEP152. However, the elongated SAS6 assemblies formed with this mutant are not procentrioles. This claim should, therefore, be toned down. Furthermore, on page 13 the authors state “the aggregative Plk4 CP mutant’s ability to potently induce procentrioles”, while on page 14 they state, “As expected, pSSTT and pS1108 epitopes and Sas6 signals were also manifest along the entire length of Plk4 CP-induced elongated procentrioles”. Again, I don’t think the authors have convincingly shown that the PLK4 CP mutant produces bone fide procentrioles at a higher frequency than wildtype PLK4. The assemblies generated by the PLK4 CP mutant are nonphysiological and should authors should be more cautious in their description.

(This is a new question came up when reviewing the revised version). A procentriole is defined as a centriole in a stage of early development, while a centriole is a cylindrical organelle composed of a set of MT triplets (Wikipedia). Our thin section TEM analysis showed that Plk4 CP mutant can induce multiple centriole-like tubular structures (Supplementary Fig. 3a,c). The new data provided in Supplementary Fig. 3b demonstrate that elongated Plk4 CP signals are decorated with acetylated tubulin, strongly suggesting that they are associated with centriolar MTs, although whether they are fully developed into normal centrioles with the nine sets of MT triplets cannot be determined. Thus, we believe that the multiple elongated Sas6 signals in Plk4 CP-expressing cells in Fig 2a,b most likely represent procentrioles or perhaps centrioles. However, to be cautious in interpreting our

findings and to accommodate this reviewer's concern, we toned down our statement by stating "procentriole-like" rather than "procentriole in lines 148, 228, and 241.

- I felt the experiment in Figure S7c was very important as it shows full-length kinase active PLK4 forms condensates in insect cells. This is critical because most of the experiments in the paper are performed in the background of the CP mutations, which may not faithfully mimic phosphorylation of the PLK4 CPB. It would be great if the authors can show that the PB2-tip mutations disrupt the PLK4 condensates formed in the insect cells. I know these experiments are time-consuming and I wouldn't insist on them for publication. However, since the PB2-tip mutations are loss-of-function mutants, it is not surprising they can overcome the gain-of-function CP phenotypes. Showing the PB2-tip mutations disrupt condensate formation in the context of full-length wildtype PLK4 would be important to interpret how the loss-of-function PB2-tip mutations affect PLK4 function.

(This is a new question came up when reviewing the revised version). The Reviewer's suggestion to generate a baculovirus-expressing Plk4 PB2-tip mutant and examine if it is condensation-defective in insect cell would be helpful. However, two lines of evidence provided in the manuscript rather strongly suggest that the PB2-tip mutations disrupt the formation of Plk4 condensates. First, in the presence of the PB2-tip mutations, the condensation-prone CPB CP mutant failed to generate *in vitro* aggregates in Fig. 6c,d. Second, while EGFP-Plk4 CP efficiently generated a dot-state morphology, its respective PB2-tip mutant exhibited an uncondensed ring-state Plk4 (Fig. 8b,c) and, consequently, failed to generate elongated procentrioles in U2OS cells (Fig. 8d). These observations strongly suggest that the Plk4 PB2-tip mutant will likely to be defective in forming condensates in insect cells. To incorporate the Reviewer #1's concern, we re-wrote that "These findings suggest that the CP PB2-tip mutant is defective specifically in its self-clustering ability----" in line 388. As the Reviewer acknowledged, the proposed baculovirus-based experiment will be time-demanding. Also, since we believe that the two lines of evidence (obtained both *in vitro* and in U2OS cells) that we already provided are strong and that the suggested experiment is not a required one for this revision, we opted out of carrying out the experiment.